

# Ocean Modeling with Adaptive REsolution (OMARE, version 1.0) – Refactoring NEMO model (version 4.0.1) with the parallel computing framework of JASMIN. Part 1: adaptive grid refinement in an idealized double-gyre case

Yan Zhang[1], Xuantong Wang[1], Yuhao Sun[2], Chenhui Ning[1], Shiming Xu[1,3], Hengbin An[4],
Dehong Tang[4], Hong Guo[4], Hao Yang[4], Ye Pu[5], Bo Jiang[2], and Bin Wang[1,5]

[1]Ministry of Education Key Laboratory for Earth System Modeling, Department of Earth System Science (DESS), Tsinghua University, Beijing, China
[2]Beihang University, Beijing, China
[3]University Corporation for Polar Research (UCPR), Beijing, China
[4]Institute of Applied Physics and Computational Mathematics (IAPCM), Beijing, China
[5]State Key Laboratory of Numerical Modeling for Atmospheric Sciences and Geophysical Fluid Dynamics (LASG), Institute of Atmospheric Physics, Chinese Academy of Sciences, Beijing, China

**Correspondence:** Shiming Xu (xusm@tsinghua.edu.cn) and Hengbin An (an_hengbin@iapcm.ac.cn)

**Abstract.** High-resolution models have become widely available to study ocean's small-scale processes. Although these models simulates more turbulent ocean dynamics and reduces uncertainties of parameterizations, they are not practical for long-term simulations, especially for climate studies. Besides scientific research, there are also growing needs from key applications for multi-resolution, flexible modeling capabilities. In this study we introduce the Ocean Modeling with Adaptive REsolution

5   (OMARE), which is based on refactoring NEMO with a parallel computing framework of JASMIN. OMARE supports adaptive mesh refinement (AMR) for the simulation of the multi-scale ocean processes with improved computability. We construct an idealized, double-gyre test case, which simulates a western-boundary current system with seasonally changing atmospheric forcings. This paper (part 1) focuses on the ocean physics simulated by OMARE at two refinement scenarios: (1) 0.5°-0.1° with static refinement and the transition from laminar to turbulent, eddy rich ocean, and (2) short-term 0.1°-0.02° AMR experiments

10  which focus on submesoscale processes. Specifically, for the first scenario, we show that the ocean kinematics on the refined, 0.1° region is sensitive to the choice of refinement region within the low-res., 0.5° basin. Furthermore, for the refinement to 0.02°, we adopt refinement criteria for AMR based on surface velocity and vorticity. Results show that temporally changing features at the ocean's mesoscale, as well as submesoscale process and its seasonality, are well captured through AMR. Related topics and future plans of OMARE, including overlaying in AMR, are further discussed for further oceanography studies and

15  applications.



## 1 Introduction

High resolution ocean models are indispensable tools for climate research and operational forecasts. Global eddy-rich models, with nominal grid resolution of $0.1°$, are capable to resolve the first baroclinic Rossby radius of radius in the mid-latitude (Chelton et al., 1998) and simulate mesoscale turbulence of the ocean (Moreton et al., 2020). Currently, these models have become common practice for both climate studies (Hirschi et al., 2020) and global ocean forecasts (Gasparin et al., 2018). With the ever-growing capability of computing facilities, global simulations at about $0.05°$ or finer have become the new frontier in recent years [Rocha et al. (2016); Chassignet and Xu (2017), among others]. Although the model's effective resolution is usually much coarser than the grid's native resolution [$5\times$ to $10\times$, see Rocha et al. (2016) and Xu et al. (2021)], the model with finer grids is capable to resolve more portion of the ocean's kinematics spectrum. Especially, the strongly ageostrophic, submesoscale processes can be partially resolved at this resolution range (D'Asaro et al., 2011). Submesoscale-rich simulations have been found to be crucially important in various findings, such as enhanced ocean heat update at ocean fronts, as well as biogeochemical impacts. Moreover, the modeled ocean energy cycles and cascading, and even the mean states are found to be better characterized at submesoscale-capable resolutions (Levy et al., 2010; Ajayi et al., 2021).

Despite its advantages, high-resolution simulations inevitably face the biggest hurdle of the daunting, even prohibitively high computational cost. Especially, long numerical integration of hundreds of years is usually required for ocean models to reach an equilibrium status. Other ensuing practice, including model parameter tuning and climate simulations, are rendered impractical for submesoscale-capable and even finer resolutions.

Given the current status and future trend of high-resolution model, there are growing need for more flexible approaches for ocean modeling. With different resolutions for different spatial/temporal locations, the model can effectively reduce the overall grid cell count hence the computations, while maintaining resolution and accuracy for key region and processes. The flexibility with the multi-resolution approach also facilitates various applications that require 'telescoping' capabilities. In the following up part of the paper, we further examine the status-quo in current models and introduce our work on adaptive refinement for ocean modeling.

### 1.1 Multi-resolution ocean modeling

Grid nesting-based regional refinement is a widely adopted approach for multi-resolution simulations. For ocean models such as Nucleus for European Modelling of the Ocean (NEMO) and Regional Ocean Modeling System (ROMS), locally refined simulations are supported through the integration with AGRIF (Debreu and Patoume, 2016). Examples include the NEMO-based, three-level grid embedding from $0.25°$ global grid to locally $1/60°$ for the study of Agulhas region (Schwarzkopf et al., 2019). At different levels of resolution, the model is configured accordingly, including the time stepping, the physics parameterization schemes and parameters. The regions with different resolutions interact through boundary exchanges, and temporal and spatial interpolations are utilized accommodate the differences in time steps and resolutions. Furthermore, AGRIF-based NEMO is adopted to construct atmosphere-ocean coupled model of FOCI (Matthes et al., 2020). Although the approach of grid nesting is friendly to existing models, there are several key issues in its current status quo. First, temporally changing, adaptive mesh/grid



refinement (AMR) have not been applied to oceanography studies, although it is widely used in traditional computational fluid
dynamics studies. Besides, reducing overall computational overhead is a major motivation for grid refinement. However, it is a
non-trivial task of to manage domain decomposition and the computational environment which are usually based on massively
parallel computers. These factors have greatly limited the model's potential to explore the multi-scale ocean processes, both
for scientific studies and key application such as operational forecasts.

Another popular approach for multi-resolution ocean modeling is to utilize non-structured grids. Examples include FESOM
(Wang et al., 2014) and MPAS (Ringler et al., 2010). With the flexibility in grid generation for non-structured grids, more
(i.e., denser) grid points can be distributed over the regions of interest. For FESOM which utilizes triangular grid cells, multi-
resolution ocean simulations are carried out, such as the regionally focused study for 'hot spots' of ocean dynamics (Sein et al.,
2016), and designing tailored grid to suite the local Rossby radius of deformation (Sein et al., 2017). Similarly, with MPAS,
grid generators that enable local refinement with Voronoi graphs (Hoch et al., 2020). Although existing models (which are
based on orthogonal grids) can no longer be utilized, this approach improves over current models in terms of higher flexibility
in modeling key regions and/or processes of the ocean. Certain limitations exist, including: (1) the model grids cannot change
arbitrarily with time, hence limited 'adaptivity' and 'flexibility', (2) scale-aware parameterization schemes should be developed
to accommodate gradual change of model grid resolution; (3) due to CFL limitations, the time step is usually controlled by the
smallest grid cell size, resulting in extra computational cost. Furthermore, there is no ocean model that utilizes non-structured
and moving grids for large-scale studies, although similar sea ice models exist such as neXtSim based on Lagrangian mesh
(Rampal et al., 2016).

In Xu et al. (2015) the authors proposed new orthogonal ocean model grids based on Schwarz-Christoffel conformal map-
pings. The new grid can redistribute grid points on the land to the ocean, with finer resolution in coastal regions. Although
the grid retains full compatibility with existing models such as NEMO, its flexibility in changing the resolution is still limited
compared with unstructured grids. Given the status quo of current ocean models, we utilize ASMIN (J parallel Adaptive Struc-
tured Mesh applications INfrastructure), a third-party, high-performance software middleware to construct a new ocean model
that support adaptive mesh refinement.

## 1.2 Flexible modeling with JASMIN

JASMIN is a parallel adaptive software framework based on C++ language, developed by Institute of Applied Physics and com-
putational mathematics (Mo et al., 2010). JASMIN aims at scientific applications based on structured grids, and the framework
supports the innovative research of physical modeling, various numerical methods and high-performance parallel environ-
ments. In particular, it facilitates the development of efficient parallel adaptive computing applications by encapsulating and
managing data and data distribution. In effect, it shields from model developers the large-scale parallel computing environment
and grid adaptivity.

For ocean models, JASMIN framework provides basic data structures, including coordinate system and grid geometry. In
particular, the support for periodic boundaries and tripolar boundaries is a natively built-in of JASMIN. For AMR, the
grid is managed through the grid hierarchy in JASMIN. The domain decomposition and mapping to parallel processes (i.e.,





Message Passing Interface, MPI) are carried out automatically by JASMIN. Furthermore, JASMIN provide other computational facilities, ranging from linear and nonliner solvers, to automatic computational performance profiling and load balancing.

In this paper, we further introduce the porting of NEMO onto JASMIN and results of OMARE with idealized, double-gyre test case. In Section 2 we introduce in detail the code refactoring process, including related model design in OMARE. Furthermore in Section 3, we test OMARE with an idealized, Double-Gyre case. Specifically, a resolution hierarchy is constructed that spans the non-turbulent to submesoscale-rich resolutions, including $0.5°$, $0.1°$ and $0.02°$. We mainly focus on the ocean physics simulated by OMARE, and a follow-up paper (part 2) will cover the computational aspects. Section 4 concludes the
paper, with a brief summary of OMARE and discussions of related topics in the development of multi-scale ocean models.

## 2   Refactoring NEMO with JASMIN

The NEMO model simulates three-dimensional ocean dynamic and thermodynamic processes governed by Primitive Equations under hydrostatic balance and Boussinesq hypothesis (Bourdallé-Badie et al., 2019). Curvilinear orthogonal, structured grids with Arakawa-C staggering are utilized in NEMO for spatial discretization and domain decomposition, as well as parallel
computation on MPI environments. Specifically, the domain decomposition is carried out in the horizontal direction (i.e., indexed by $i$ and $j$ respectively). Various parameterization schemes are available for sub-grid scale processes, including first-order and second-order viscosity/diffusion for lateral mixing and turbulent-closure models for vertical mixing. In its current implementation, NEMO is based on FORTRAN, with all model variables defined and accessed as global variables in various modules, including grid variables, prognostic variables, etc. Furthermore, the decomposition in NEMO (as well as AGRIF-
based NEMO) is currently based on predefined block sizes and cannot change during the time integration. In order to enable adaptive refinement in NEMO, we need more flexibility through the support of dynamically changing grids and the ensuing grid decomposition.

    As a third party software middleware, JASMIN provides scientific applications the adaptive mesh refinement through another abstraction layer of structured grids and grid decomposition. Besides, JASMIN also shields the computational aspects, including message passing, input/output, from developers. In order to utilize these functionalities, we need to refactor the code-
base of NEMO onto JASMIN, following JASMIN's routines. Since NEMO is based on FORTRAN, the refactorization onto JASMIN also involves FORTRAN/C++ hybrid programming. Key terms of JASMIN's nomenclature are as follows.

- ***Grid hierarchy***: a series of (recursively) embedding grids with several resolution levels.

- ***Patch***: a basic rectangular (i.e., two-dimensional) region with generic sizes, which holds a certain variable of a given
depth.

- ***Integrator Component*** (or ***Component*** for short): a basic unit for time integration, consisting of a boundary exchange of a certain variable set of a patches, followed by a series of computation on the patches. The whole time integration consists of the function calls of a series of components.





The refactorization process involves two aspects: (1) all the data in NEMO (grid variables, prognostic variables, etc.) are

transferred and managed by JASMIN, (2) all the code in NEMO (dynamic core, parameterization schemes) are reformulated into JASMIN components. Finally, we need to rewrite the whole time integration in C++, which consists of calls to various components. The time step is iteratively called by JASMIN for time integration. As compared with AGRIF-based NEMO, the refactorization with JASMIN consists of much larger overhaul of the codebase, since all the data and communication are managed through JASMIN. Details of the refactorization process is introduced in detail below.

**2.1 Code refactoring strategy**

In NEMO, the time integration process is divided into subroutines distributed in various FORTRAN modules. In order to match the scheme of 'communication-compute' of JASMIN components, subroutines with several communication calls need to be further segmented. Furthermore, the variables which are used by the subroutines and reside in global spaces in NEMO, need to provided through the component interfaces from JASMIN patches. Therefore, in order to ensure correctness, we adopt a

bottom-up strategy to conduct code refactorization process, including three steps: (1) the separation of communication, (2) the standardization of call interfaces, and (3) the formulation of JASMIN components. The whole process is shown in Figure 1, with each step detailed in Section 2.1.1 through 2.1.3. In total, during the refactorization we have formulated 155 components and 422 patches in OMARE.

### 2.1.1 Communication separation

We separate the MPI communication in NEMO (e.g., `lbc_lnk` and `mpp_sum`) from subroutines and divide each subroutine into smaller ones which only contain computational codes. As in Figure 3, each step of the time integration of NEMO consists of a series calls to FORTRAN subroutines (step 0). For the first step of separating the communication, we segment the core computing subroutines from the calls for boundary exchanges. In Figure 3, subroutine 1 shares the same input as the core computing subroutine 1.1 and the same output as the core computing subroutine 1.2, with a communication in between. Taking

the subroutine `dyn_adv` as an example, we first expose the 'select-case' structure in the time-stepping program for more direct control of the parameterization schemes. Notice that there is a required boundary exchange (`lbc_lnk`) within the subroutine `dyn_keg`, we separate it and split `dyn_keg` into two core computing subroutines: `dyn_keg1` and `dyn_keg2`. For comparison, since the subroutine `dyn_zad` does not contain communication, it consists a single computing subroutine. Consequently, for `dyn_adv` we finish the communication separation and get three core computing subroutines (Step 1 of

Listing 1).

### 2.1.2 Standardization of interfaces

NEMO manages all variables in the public workspace in various modules, so that every subroutine can directly access these variables without the need to passing them as parameters. This simplifies the coding process, but compromises the 'stateless-ness' of the code. Here, we complement the argument list of every core computing subroutine with a standardized interface





**Figure 1.** Overview of code refactoring process.

(step 2 of Figure 1). The standardized interface contains all variables the subroutine needs and divides those arguments into four categories: `time`, `index`, `field` and `scalar`. This process makes the subroutine 'stateless'. The `time` refers to the time step (a.k.a, `kt` in NEMO). The `index` refers to the generic size information of domain decomposition, such as `jpi`, `jpj` and `jpk`. The `field` refers to field variables, such as prognostic variables, diagnostic variables and additional scratch-type variables. They corresponds to data held by JASMIN patches. The `scalar` refers to parameters to the process control in the subroutine. All these arguments are declared and defined in the newly, standardized version the subroutine. As an example, for core computing subroutine `dyn_keg1`, the original subroutine `dyn_keg` only contains the time step argument `kt` and a scalar argument `kscheme` (Listing 2.a). After complementing the argument list during standardizing the interface (Listing 2.b), the arguments are organized by four categories. Besides `kt` and `kscheme`, the `index` parameters–`jpi`, `jpj`, `jpk`, `jpim1`, `jpjm1`, `jpkm1` and field variables `ua`, `va`, `un`, `vn` which are from NEMO's *oce* module are included. One thing to note is: we change three local variables to the public space of the module *dynkeg* and rename them by the suffix of module name, in case that they need to be transferred between the splitted subroutines. In the front of subroutine `dyn_keg1`, we





**Listing 1** Code example of subroutine `dyn_adv` from communication separation to interface standardization. The highlight area are actually excuted code in the porting example.

| Step 0. Original program | Step 1. Communication sepration | Step 2. Interface standardization |
|---|---|---|

```fortran
SUBROUTINE stp( kstp )
.
.
.
.
.
.
.
.
.
.
.
.
.
.
.
.
 CALL dyn_adv( kstp )
.
.
.
.
.
.
.
.
.
.
.
.
.
END SUBROUTINE stp
```

```fortran
SUBROUTINE stp( kstp )
.
.
.
.
.
.
.
.
.
.
.
.
 SELECT CASE( n_dynadv )
 CASE( np_VEC_c2 )
   CALL dyn_keg1 ( kstp, nn_dynkeg )
   CALL lbc_lnk  ('dynkeg', zhke, 'T', 1.)
   CALL dyn_keg2 ( kstp )
   CALL dyn_zad  ( kstp )
 CASE( np_FLX_c2 )
   ! CALL dyn_adv_cen2( kstp )
 CASE( np_FLX_ubs )
   ! CALL dyn_adv_ubs( kstp )
 END SELECT
.
.
.
.
.
.
.
.
.
END SUBROUTINE stp
```

```fortran
SUBROUTINE stp( kstp )
.
.
.
 SELECT CASE( n_dynadv )
 CASE( np_VEC_c2 )
   CALL dyn_keg1 ( kstp,                      &
     jpi, jpj, jpk, jpim1, jpjm1, jpkm1,      &
     vua, va, un, vn,                         &
     zhke_DYNKEG, ztrdu_DYNKEG, ztrdv_DYNKEG, &
     nn_dynkeg )
   CALL lbc_lnk  ('dynkeg', zhke_DYNKEG, 'T', 1.)
   CALL dyn_keg2 ( kstp,                      &
     jpi, jpj, jpk, jpim1, jpjm1, jpkm1,      &
     ua, va, e1u, e2v, umask, vmask,          &
     zhke_DYNKEG, ztrdu_DYNKEG, ztrdv_DYNKEG)
   CALL dyn_zad  ( kstp,                      &
     jpi, jpj, jpk, jpim1, jpjm1, jpkm1,      &
     ua, va, un, vn, wn,                      &
     e1e2t, r1_e1e2u, r1_e1e2v, e3u_n, e3v_n, &
     umask, vmask)
 CASE( np_FLX_c2 )
   ! CALL dyn_adv_cen2( kstp )
 CASE( np_FLX_ubs )
   ! CALL dyn_adv_ubs( kstp )
 END SELECT
.
.
.
END SUBROUTINE stp
```

declare the data type, size and the intent type of each argument and arrange them according to their source modules. Taking the same way to reconstruct subroutine `dyn_keg2` and `dyn_zad`, we can finally refactor the subroutine `dyn_adv` from Step 1 to Step 2 in Listing 1. After the standardization of interfaces, the whole code base is still based on NEMO. There is no change in result, which is used to correctness check of the refactorization process.

### 2.1.3 Formation of JASMIN components

Next we change the whole NEMO environment to JASMIN. The control of the MPI is to be transferred from NEMO to JASMIN, including the initialization of MPI environment, domain decomposition, process mapping and communication. Besides, all NEMO variables need to be replaced as the patch data which are both initialized and provided by JASMIN.





**Listing 2** Code example of subroutine `dyn_keg1` for the interface standardization.

a. Before interface standardization

```fortran
SUBROUTINE dyn_keg( kt, kscheme )
   INTEGER, INTENT( in ) ::  kt
   INTEGER, INTENT( in ) ::  kscheme
   !
   INTEGER ::  ji, jj, jk
   REAL(wp) ::  zu, zv
   REAL(wp), DIMENSION(jpi,jpj,jpk)        ::  zhke
   REAL(wp), ALLOCATABLE, DIMENSION(:,:,:) ::  ztrdu, ztrdv
   .
   ! numerical computation
   .
END SUBROUTINE dyn_keg
```

b. After interface standardization

```fortran
SUBROUTINE dyn_keg1(kt,                              &
        jpi, jpj, jpk, jpim1, jpjm1, jpkm1,      &
        ua, va, un, vn,                          &
        zhke_DYNKEG, ztrdu_DYNKEG, ztrdv_DYNKEG, &
        kscheme)
   !!--------self---------!!
   INTEGER, INTENT(in) ::  kt
   INTEGER, INTENT(in) ::  kscheme
   !!--------external---------!!
   !mod oce
   REAL(wp), DIMENSION(jpi,jpj,jpk), INTENT(inout)   ::  ua, va
   REAL(wp), DIMENSION(jpi,jpj,jpk), INTENT(in)      ::  un, vn
   !mod par_oce
   INTEGER, INTENT(in)  ::  jpi, jpj, jpk
   INTEGER, INTENT(in)  ::  jpim1, jpjm1, jpkm1
   !local transfer
   REAL(wp), DIMENSION(jpi,jpj,jpk), INTENT(inout) :: zhke_DYNKEG, ztrdu_DYNKEG, ztrdv_DYNKEG
   !!--------local-------!!
   INTEGER ::  ji, jj, jk
   REAL(wp) ::  zu, zv
   .
   ! numerical computation
   .
END SUBROUTINE dyn_keg1
```





---

**Listing 3** Code example of subroutine `dyn_adv` in JASMIN time-stepping arrangement after componentization.

Step 3. JASMIN componentization

```
Class DynKeg2PatchStrategy{
    .
    .
    .
    void initializeComponent(...){
        // MPI communication
        intc->registerRefinePatchData(manager->zhke_DYNKEG_id,
                                      manager->zhke_DYNKEG_id,
                                      "NEMO_BL_INTERP");
    }
    .
    .
    .
    void computeOnPatch(...){
        // get patch data from JASMIN
        .
        .
        // index transform from JASMIN to NEMO
        .
        .
        __dynkeg_MOD_dyn_keg2(kt,
            jpi, jpj, jpk, jpim1, jpjm1, jpkm1,
            ua->getPointer(), va->getPointer(), e1u->getPointer(), e2v->getPointer(),
            umask->getPointer(), vmask->getPointer(),
            zhke_DYNKEG->getPointer(), ztrdu_DYNKEG->getPointer(), ztrdv_DYNKEG->getPointer());
    }
    .
    .
    .
};
```

---

For each subroutine with the standardized interface, we provide a JASMIN wrapper which is a derived C++ class of JASMIN component. For the exemplary subroutine `dyn_keg2` with standardized interface, the component `DynKeg2PatchStrategy` is constructed using JASMIN nomenclature (Listing 3). Furthermore, two functions are implemented (i.e., instantiated virtual functions in C++). Firstly, the function `initializeComponent` serves as the necessary boundary exchange for the patch data `zhke_DYNKEG`, which corresponds to the FORTRAN subroutine `lbc_lnk` in NEMO. The argument "`NEMO_BL_INTERP`"

is a user-defined spatial interpolator during refinement, and it will be further discussed in Section 2.2. Secondly, we implement the function `computeOnPatch` which is a wrapper to the FORTRAN subroutine `dyn_keg2`. On the technical side, it is actually renamed as `__dynkeg_MOD_dyn_keg2` after name mangling through FORTRAN/C++ hybrid compilation. Furthermore, we need to get patch data from JASMIN workspace and transform JASMIN index-related variables to the subroutine's interface, so that to complete the argument list. This component patch strategy will be called by a corresponding numerical

integrator component `DynKeg2_intc`, thus can be called by the time step in JASMIN (step 3 of Figure 3).





---

**Listing 4** Code example of subroutine `dyn_adv` in JASMIN time-stepping program after formulating the components. The highlight area are the actually executed code in the porting example.

---

Step 3. JASMIN time-stepping program

```cpp
int NemoLevelIntegrator::advanceLevel(...){
  .
  .
  .
  /*=====================*
   *       dyn_adv       *
   *=====================*/
  int n_dynadv;
  const int np_VEC_c2 = 1;
  const int np_FLX_c2 = 2;
  const int np_FLX_ubs = 3;
  n_dynadv_nemo2jasmin_(n_dynadv);
  switch (n_dynadv)
  {
    case( np_VEC_c2 ):
        DynKeg1_intc->computing(level, current_time, predict_dt);
        DynKeg2_intc->computing(level, current_time, predict_dt);
        DynZad_intc->computing(level, current_time, predict_dt);
        break;
    case( np_FLX_c2 ):
        tbox::pout << "CALL dyn_adv_cen2" << endl;
        break;
    case( np_FLX_ubs ):
        tbox::pout << "CALL dyn_adv_ubs" << endl;
        break;
  }
  .
  .
  .
};
```

Listing 4 shows the porting result of the whole subroutine `dyn_adv` which is already in C++. The time-stepping function `advanceLevel` serves to do the integration in JASMIN like the subroutine `stp` in NEMO. The JASMIN (or OMARE) version of subroutine `dyn_adv`, which is part of `stp`, is composed of three numerical integrator components: `DynKeg1_intc`, `DynKeg2_intc` and `DynZad_intc`. The actual operations of these components are carried out by calls to the function of
`computing`. Three variables are transferred: `level` refers to the current level, `current_time` refers to the current time steps, `predict_dt` refers to the timestep in the current level. The switch-case structure in C++ is consistent with the logic of the original FORTRAN code in NEMO. The function calls to the other two parameterization schemes are put in place for future porting.

Following this manner, we refactor the whole time integration of NEMO onto JASMIN (step 3 of Figure 1). Under the
JASMIN context, the integrator component frees users from parallel programming and further supports AMR. For reductions





for global diagnostics in NEMO, we utilize JASMIN's reduction integrator components. For model initialization, we also design dedicated initialization integrator component, which manages all field variables through JASMIN patches. The allocation, deallocation, and communications are then carried out and further managed by JASMIN. Since the patches are all managed by JASMIN, including their distribution on processors, the operations required for AMR are enabled by JASMIN, including the change of refinement settings, patch generation, re-doing of domain decomposition into patches, etc.

## 2.2 Refinement in OMARE

OMARE utilizes the two-dimensional adaptive refinement functionalities of JASMIN. The spatial refinement is only carried out in the horizontal directions (similar to AGRIF-based NEMO), mainly due to the anisotrophy between the horizontal and vertical directions of the ocean processes at large scales. Multi-level, recursive refinement is also supported, with the refinement ratio between the resolution levels specified by users in the OMARE's namelist. Accordingly, temporal refinement is accompanied with spatial refinement, and it can be set independently from spatial refinement ratio, in order to be flexible for improved efficiency and stability. Furthermore, in current version of OMARE, we only consider one-way, coarse-to-fine forcings, but not the interaction or feedback from the fine level. Related issues are discussed in Section 3 and further in Section 4. Details of the refinement in OMARE are introduced below.

### 2.2.1 Time integration

To support adaptive refinement in OMARE, we introduce *super cycles* in the time integration. Each super cycle consists of a fixed number of baroclinic steps at the coarsest resolution of the grid hierarchy. At the beginning of each super cycle, the grid refinement setting can be adjusted by specifying a refinement map (also a *patch* in JASMIN). The refinement map contains the Boolean flags marking each grid point whether to be refined to the next level of resolution. User-specified rules can be integrated in OMARE, including adaptivity to temporally changing features (details in Sec. 3.4). The time step for the super cycle (i.e., baroclinic step count) can also be specified in the namelist of OMARE.

Figure 2 shows the overall time integration in OMARE. After the refinement map is set at the beginning of the super cycle, the grid hierarchy will be (re)constructed if the map is changed. The construction consists of the domain decomposition, the mapping to the MPI processes, the construction of the communication framework, as well as necessary operations on the model status. First, the migration of model status is needed in the case of a new domain decomposition. The second case is to create fine-resolution model status from that on coarse grid, for newly refined regions. The level hierarchy is a data structure that manages all levels of the JASMIN framework, including the level number, refined ratio, etc. In OMARE, currently we initialize the model's prognostic status on each newly-created fine-resolution grid point with surrounding coarse-resolution grid points through (bi-)linear interpolations.

For each baroclinic step (on the coarse level), the time integration is carried out recursively on all levels throughout the grid hierarchy. The temporal refinement ratio controls the baroclinic step count on the finer levels. On the finer level, the time integration is carried out after the integration on the coarser level is finished (i.e., time step $k+1$ in Fig. 3). Accordingly, on the lateral boundaries between the coarse and fine levels, the model status on the coarse level provide these conditions at each fine-




level time step (between $k$ and $k+1$ step on the coarse level in Fig. 3). For OMARE, since we use the split-explicit formulation
for barotropic and baroclinic processes with each baroclinic step contains several barotropic steps, and model status for the
barotropic and baroclinic steps are kept on the coarse level grid for spatial and temporal interpolations for the fine level.

### 2.2.2 Inter-level interpolations

In order to force the fine level with the the outer, coarse level, an inter-level halo region is automatically created for the fine
level by JASMIN. The halo region has the same resolution as the fine level, but the model status are specified on-fly with the
coarse level model status. The width of the halo region can be specified through OMARE's namelist. Figure 3 shows a sample
case involving two resolution levels, with: (1) the (spatial) refinement ratio of 3, and (2) the halo width of 1. In particular, the
refined region has a irregular shape (i.e., non-square), and the inter-level halo is created accordingly (marked in yellow). The
physical boundary of the case, marked in grey, does not participate in the inter-level halo but constrains the model status on
both the coarse and the fine levels through the physical boundary conditions.

For the spatial and temporal interpolation of the model status on the inter-level halo, we currently implement a basic set
of linear interpolations in OMARE. For temporal interpolation, we adopt intrinsic interpolators built in JASMIN, which is
standard linear interpolation. This applies to both barotropic and baroclinic model variables as provided on the coarse level. For
spatial interpolation, we implement variable-specific (bi)-linear interpolators. Specifically, we implement three interpolators:
`NEMO_BL_INTERP`, `NEMO_U_INTERP` and `NEMO_V_INTERP`, which are designed for non-staggered variables, variables
on the east edge (i.e., $u$ for Arakawa-C grid), and those on the north edge (i.e., $v$). As shown in Figure 3, the data on non-
staggered locations of the target fine-level cell $R$ is attained by interpolating the data of 4 surrounding coarse-level cells ($C_1$
to $C_4$), which is implemented in `NEMO_BL_INTERP`. For $u$-points, the interpolation involves two or four adjacent cells. The
data $U_{c1}$ to $U_{c4}$ on the coarse $t$-point $C_1$ to $C_4$ corresponds to the $u$-point $C_1' \sim C_4'$ (from the dashed vector to the solid vector
in Figure 3), with its location at $R'$. This interpolator is implemented as `NEMO_U_INTERP`. Similar treatment to points on
$v$-points in Arakawa-C grid is carried out accordingly, which corresponds to `NEMO_V_INTERP`. For the fine cells near the
physical boundary, we have the following specific treatments. In the case of lack of valid data for interpolation, the bilinear
interpolation degenerates to the linear or nearest-neighbor interpolation. For model status on physical boundaries, we override
the interpolated values according to proper boundary conditions if necessary.

In general, the overall routines of inter-level interpolation are similar to AGRIF-based NEMO, with the key difference in
OMARE that irregular shape of refinement, as well as adaptive refinement are enabled in OMARE. User-specified interpolators,
including `NEMO_BL_INTERP`, `NEMO_U_INTERP` and `NEMO_V_INTERP`, are examples of bespoke interpolation functions
that are possible in JASMIN. More sophisticated interpolators [i.e., conserved ones for fluxes, high-order algorithms, as in
Debreu et al. (2012) and AGRIF-based NEMO] are planned for future development of OMARE.





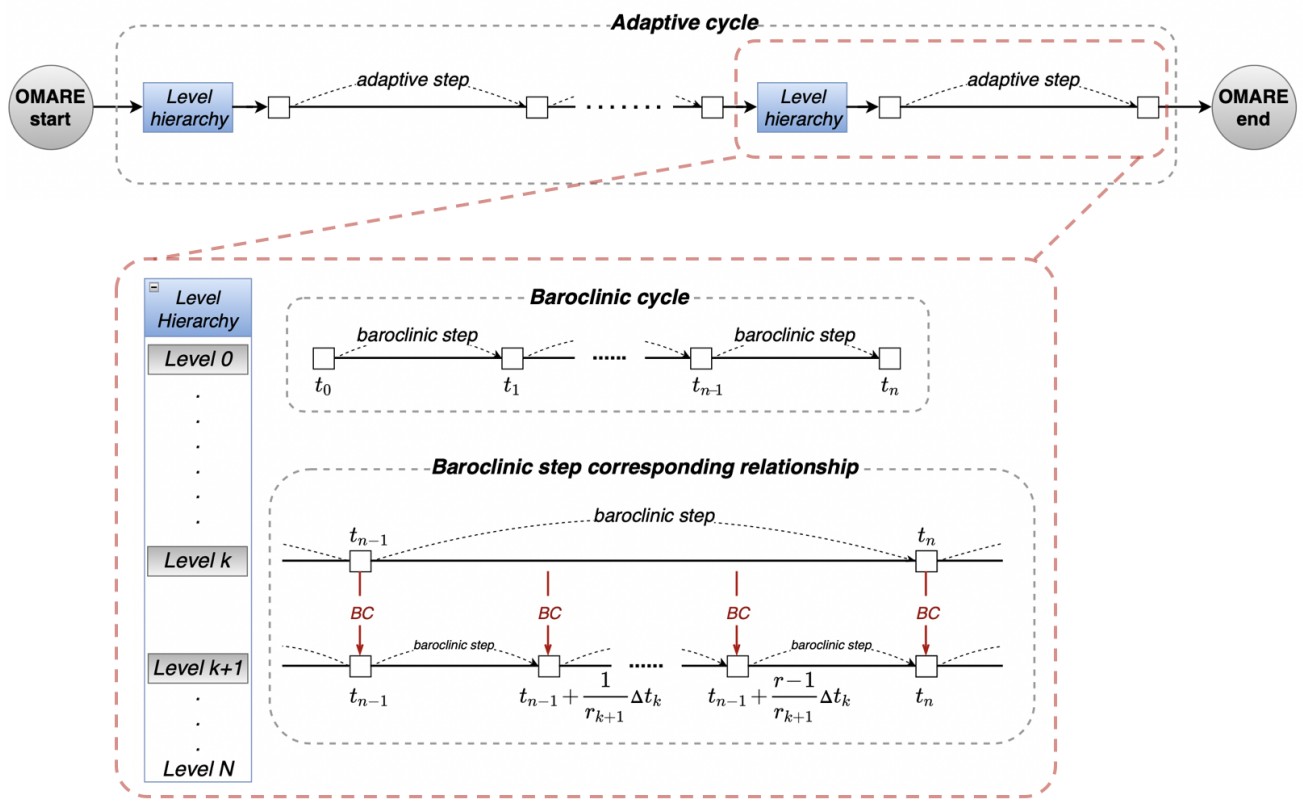

**Figure 2.** Flow chart of the adaptive framework in OMARE.

## 3 Model settings and numerical experiments

### 3.1 Typical resolutions and model configurations

In OMARE we focus on three typical spatial resolutions, as shown in Table 1. While $0.5°$ resolution (or coarser) is mainly used for climate models and long-term time integration, the model cannot simulate the ocean's mesoscale process and the geostrophic turbulence. We denote the ocean as simulated by $0.5°$ model as laminar ocean. The second resolution of $0.1°$ is 5 times finer than the first resolution of $0.5°$, and is commonly used in the community for eddy/mesoscale-rich simulations. The nominal $10\text{-}km$ grid spacing is capable to resolve the first baroclinic Rossby radius of deformation in mid-latitudes, which is about $50\ km$ at $30°N$ (Chelton et al., 1998). The finest resolution is $0.02°$ and another 5 times finer that $0.1°$. This resolution corresponds to about $2\ km$ grid spacing in the midlatitudes, and it is usually adopted for submesoscale-rich simulations (Rocha et al., 2016). These resolutions form the three-level resolution hierarchy in OMARE .

Furthermore, we adopt the following model configurations for each specific resolution. The baroclinic time steps of the three resolutions are chosen as 1 hour for $0.5°$, 600 seconds for $0.1°$ (or 1/6 that of $0.5°$), and 120 seconds for $0.02°$ (or 1/5



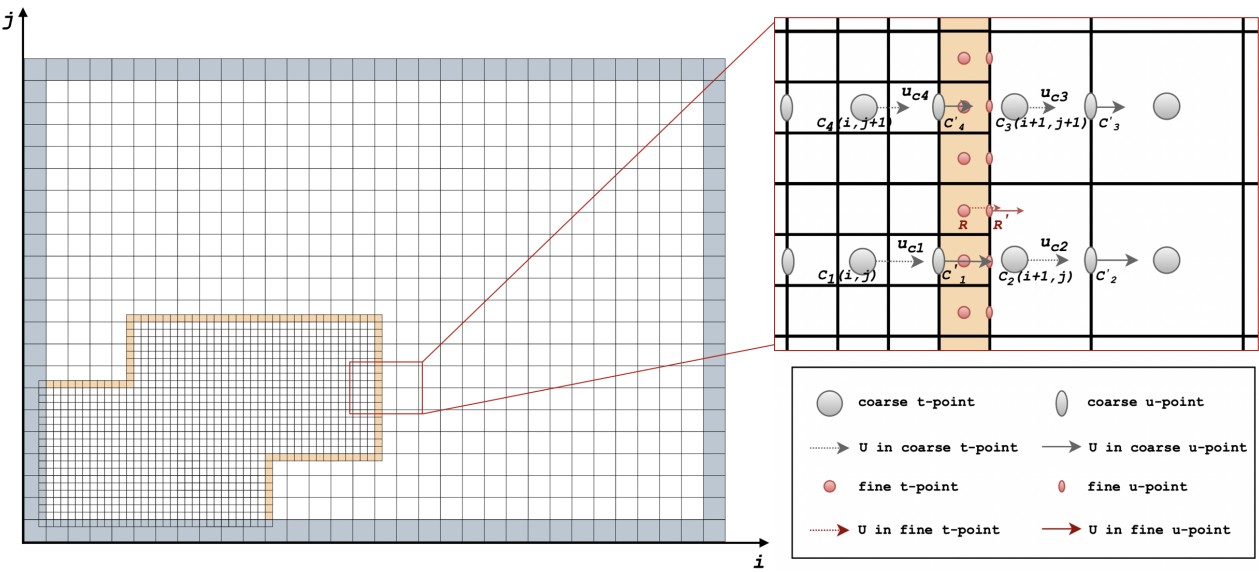

**Figure 3.** Example of a refinement region with an irregular boundary (refinement ratio 1:3) with the idealized test case in Sec. 3. The size of the coarse grid for the modeled ocean basin is $30 \times 20$. Physical boundaries are marked by grey cells. The inter-level halo, marked by yellow cells, is controlled by the coarse level, but on the same resolution as the fine level. A specific region on the coarse level-fine level boundary is shown on the right, with details of the variable layout ($u$ and $t$).

that of $0.1°$), respectively. The purposeful decrease ratio in time step at $0.1°$ of $1/6$ (instead of the 1:5 resolution difference) is to ensure better numerical stability at finer resolutions. The barotropic time step is computed proportionally according to the baroclinic time step size, in order to accommodate the nominal surface gravity wave speed. For the vertical coordinate, OMARE uses the same $z$-coordinate, with the same vertical layers across the three horizontal resolutions. The (adaptive) grid

refinement is only carried out in the horizontal direction. For the vertical coordinate, we adopt the following scheme for all numerical experiments: the layer depth starts at $8\ m$ in the mixed layer, and gradually increases to over $150\ m$ towards the ocean abyss. For the maximum depth of $4200\ m$, there are in total 50 vertical layers.

For the horizontal mixing parameterization, we adopt different schemes for the three resolutions. For momentum mixing, at $0.5°$ we adopt the first-order Laplacian isotropic viscosity, and at $0.1°$ and $0.02°$, we adopt second-order, bi-Laplacian

viscosity. For tracer mixing, we simply apply an uniform diffusivity parameter across the resolutions (Tab. 1). For the vertical mixing parameterization, we adopt the same turbulent closure parameterization scheme (TKE) across the three resolutions, with enhanced mixing for very weak stratification. The choice of parameters is preliminary and subjected to further tuning in the future.

 

**Table 1.** Model configurations of the resolution hierarchy of three levels.

| Level | Characteristics | Resolution | Time step | Horizontal mixing | Vertical mixing |
|---|---|---|---|---|---|
| 1 | Laminar ocean | $0.5°$ | $3600\ s$ | Momentum: Laplacian $K_m = 2.5 \times 10^5 m^2/s$ | |
| | | | | Tracer: Laplacian $K_t = 2.5 \times 10^2 m^2/s$ | TKE |
| 2 | Mesoscale-rich ocean | $0.1°$ | $600\ s$ | Momentum: Bi-Laplacian $K_m = 10^{10} m^4/s$ | $(100 m^2/s$ |
| | | | | Tracer: Laplacian $K_t = 2.5 \times 10^2 m^2/s$ | for $K_{evd})$ |
| 3 | Submesoscale-rich ocean | $0.02°$ | $100\ s$ | Momentum: Bi-Laplacian $K_m = 8 \times 10^7 m^4/s$ | |
| | | | | Tracer: Laplacian $K_t = 2.5 \times 10^2 m^2/s$ | |

## 3.2 Double-Gyre test case

In this study we use an idealized Double-Gyre test case to test OMARE with the mesoscale and submesoscale processes on the western boundary current (WBC) system. The modeled region is a rectangular, closed ocean basin on the $\beta$-plane centered at 30°N. The size of the ocean basin is 3000 $km$ in the zonal direction, and 2000 $km$ in the meridional direction. The depth of the ocean basin is uniformly 4200 $m$ (or 50 layers). Free-slip lateral boundary condition is used for all the experiments and across all three resolutions.

The atmospheric forcing is a normal-year, seasonally changing forcing with both dynamic and thermodynamic components. Each model year consists of 360 days, divided into 12 months with 30 days per month. The wind stress is purely zonal (i.e., only $U$-wind stress), and a distinct seasonal cycle of both wind strength and wind direction turnaround latitude. The thermodynamic atmospheric forcing is carried out by using a temperature-based recovery condition for the ocean surface. The forcing is introduced in detail in Appendix A, with Figure 4 showing the extreme conditions in summer and winter.

The model is initialized to a stationary state with uniform vertical profiles for temperature and salinity across the basin. The surface ocean kinetics reaches a quasi-equilibrium status after 20 years from the start, forming the sub-tropical gyre, the sub-polar gyre, as well as the western boundary current (WBC) system. Figure 4.c shows the annual mean sea surface height (SSH) after 50 years with 0.5° resolution.

## 3.3 From laminar ocean to mesoscale-rich, turbulent ocean

Based on 50-year spin-up run with 0.5° resolution, we carry out three experiments with 0.1° resolution. We denote the experiment with 0.5° resolution as **L**, standing for 'laminar ocean'. A full-field 0.1° experiment (denoted **M**) is carried out through the online refinement based on 0.5° experiment by mapping of the model's prognostic status on 0:00, Jan-1 at model year 51. Furthermore, two parallel 0.5°-0.1° experiments with different regional refinement to 0.1° is carried out, as shown in Figure 5. Since these two experiments involve two interactive resolutions, we denote them as **L-M-I** and **L-M-II**, respectively. The region of 0.1° resolution is interconnected and has irregular boundaries, covering the western boundary, the majority of the northern and southern boundary, WBC and its extension. Besides, **L-M-I** covering more (less) area in the subtropical (subpolar) gyre than **L-M-II**. The region of 0.1° are 56% and 61% of the whole basin for **L-M-I** and **L-M-II**, respectively.



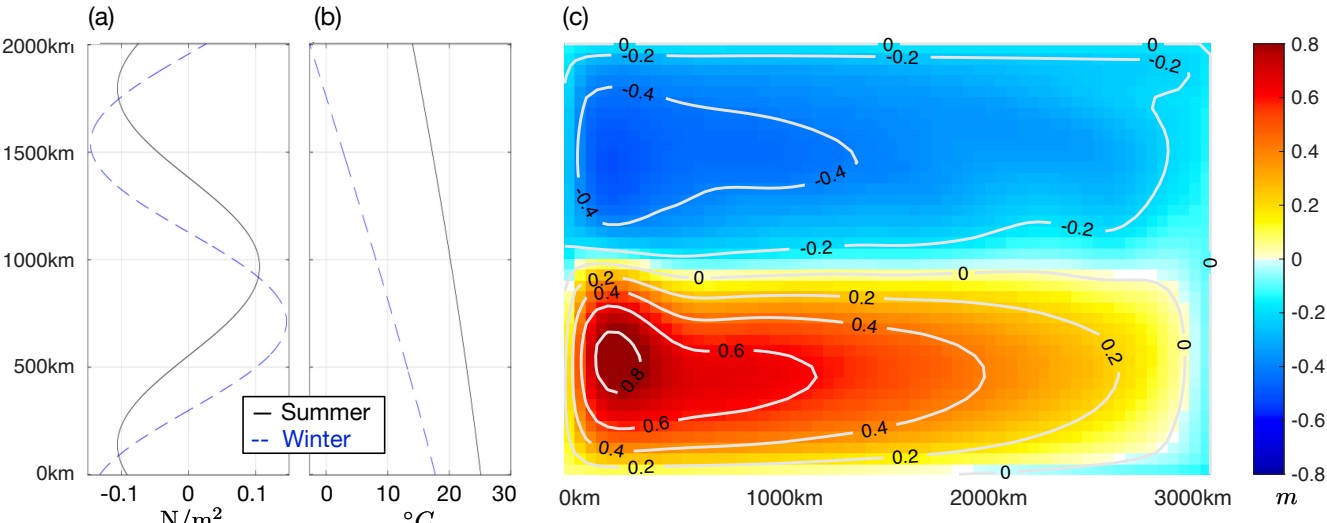

**Figure 4.** Zonal wind stress (a), sea-surface temperature (b), and the sea-surface height climatology under $0.5°$ resolution (c).

Compared with $0.5°$ resolution, the experiment with $0.1°$ resolution shows much higher overall kinetic energy (Fig. 6). At equilibrium, the annual mean surface KE is about $0.035\ m^2/s^2$ for $0.1°$, which is about $2.5\times$ that at $0.5°$. This corresponds to
the mesoscale turbulence which manifests at the spatial resolution of $0.1°$ ($10\ km$) or finer. For comparison, at $0.5°$ the model only simulates laminar flow, despite the presence of WBC (Fig. 4.b and video supplementaries). Also the distinct seasonal cycle of surface KE at $0.1°$ is both larger in magnitude ($0.01\ m^2/s^2$ v.s. $0.005\ m^2/s^2$) and different in terms of maximum month (May v.s. March).

Interestingly, there is a sudden jump of KE (from $0.015\ m^2/s^2$ to $0.07\ m^2/s^2$) within the first 3 months of the resolution
change to full-field $0.1°$. The excessively high KE gradually decrease towards the end of year 53 (i.e., 3 years after the change in the resolution). Further examination of the SSH field at equilibrium status for $0.1°$ experiment reveals systematically lower PE than $0.5°$ experiment (Fig. 7, first row). Both the subtropical gyre and the subpolar gyre show lower absolute values of SSH for $0.1°$ experiments (Fig. 7.b). Correspondingly, while the SSH difference between the two gyre centers is over $1.2\ m$ for $0.5°$ experiment, that for $0.1°$ experiment is only about $70\ cm$. This results indicate that there is higher potential energy (PE)
in the $0.5°$ experiment that cannot be sustained in $0.1°$ experiment. The PE is released after the change of resolution to $0.1°$, causing the high KE during the first 2 years after resolution change. The overall difference in the kinetics and PE, as well as the transitional phase, imply the drastically different energy cycles at the two resolutions. While $0.5°$ or similar resolutions are adopted in climate models, it cannot accurately characterize the conversion of energy from the atmospheric kinematic input and the ensuing energy transfer and cascading.

Furthermore, the two regional refinement experiments (**L-M-I** and **L-M-II**) reveal further evidence of the energy cycles. Similar to full-field $0.1°$ experiment, they both show: (1) a high KE during the first two years (i.e., a *spin-up* process) after



refinement, and (2) a gradual adjustment to equilibrium KE which is attained after 5 years. However, although they both have partial region of the basin with 0.1° (56% and 61% respectively), they show higher KE than the full-field 0.1° experiment. Especially, for **L-M-I**, the mean surface KE is at 0.054 $m^2/s^2$ (year 56 to 60) which is about 50% higher than **M**. For **L-M-II**,

the mean surface KE is marginally higher than **M** by 14 %, at 0.04 $m^2/s^2$.

The systematically higher KE is mainly due to the lateral forcing of 0.5° on refined, 0.1° regions through the boundary (Fig. 7). In both **L-M-I** and **L-M-II**, the region of 0.1° contains the WBC, the western boundary of the basin, and the majority of the northern and southern boundary. The full-field 0.5° simulation, which is carried out online with the 0.1° simulation, casts influence on the 0.1° region through the boundary, affecting all the model's prognostic status including barotropic speed,

baroclinic speed, surface height, temperature and salinity. The effect of the systematically high SSH in 0.5° region is most evident on the zonal 0.5°-0.1° boundary in the subtropical gyre in **L-M-I**. There is a local SSH maximum (indicated by 0.6-$m$ SSH isobath) which is separated from the SSH core of the subtropical gyre to the west end of the basin. For comparison, in **L-M-II** the region with 0.1° in the subtropical gyre extends further to the east by 800 $km$. As a result, the SSH in the subtropical gyre, including both the SSH on the 0.5°-0.1° boundary and the overall SSH of the gyre, is reduced and more consistent with

the full-field 0.1° experiment (**M**).

The modeled SSH and surface KE also shows higher sensitivity to the refinement in the subtropical gyre than in the subpolar gyre. Comparing **L-M-I** and **L-M-II**, we show that with increased (reduced) zonal coverage of the refined region for both subtropical and subpolar gyre, the mean SSH in the gyre becomes closer to that in **M**. However, when more portion of the subtropical gyre (other than the subpolar gyre) is included for refinement, both SSH and KE shows a higher degree of

consistency with the full-field 0.1° experiment. This is potentially related to the higher absolute SSH in the subtropical gyre than the subpolar gyre, which is consistent across both 0.5° and 0.1° experiments. With higher SSH in the subtropical gyre as simulated with 0.5°, the boundary between 0.5° and 0.1° enforces higher SSH in the 0.1° region, with more ensuing PE-KE conversion and higher KE (Fig. 6). In summary, we find that the difference in sensitivity to refinement is the result of both the climatological SSH, and the specific choices of refinement regions. Although eddy-rich ocean simulations are becoming more

available to the community, climate simulations still rely heavily on 0.5° or even coarser resolutions (Tsujino et al., 2020). The resolution hierarchy of 0.5°, 0.1° and 0.5°-0.1° refinements in OMARE serve as a basic framework for the further study of the oceanic energy cycle of numerically laminar and turbulent oceans.

### 3.4 Adaptive refinement to 0.02° and submesoscale processes on WBC

Based on the full-field 0.1° experiment, we further carry out the refinement to 0.02°, and focus on submesoscale processes

on the WBC. Specifically, we adopt adaptive refinements in order to capture temporally varying mesoscale and submesoscale features, using surface velocity and relative vorticity as proxies. The model dynamically determines the region of refinement to 0.02° based on instantaneous flow fields, based on threshold values as listed in Table 2. If the absolute value at each grid cell is larger than the threshold, the cell is marked for refinement. The adaptive refinement is achieved by recomputing the region of refinement at a fixed interval (i.e., the super cycle of 5 model days).





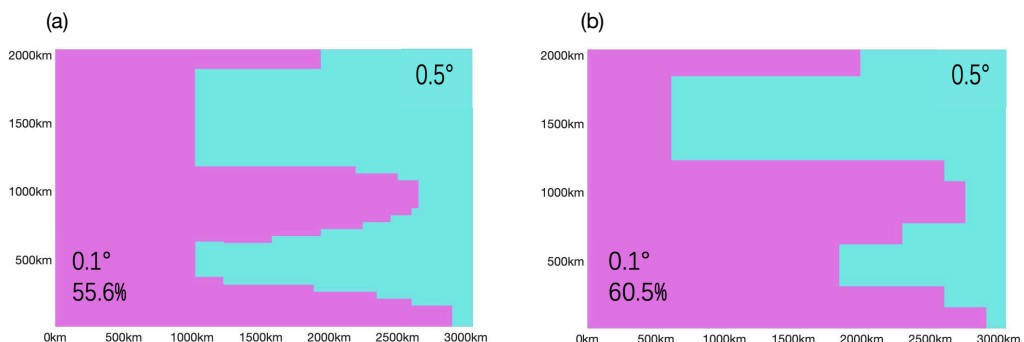

**Figure 5.** Two regional refinement settings from $0.5°$ to $0.1°$ of the Double-Gyre case.

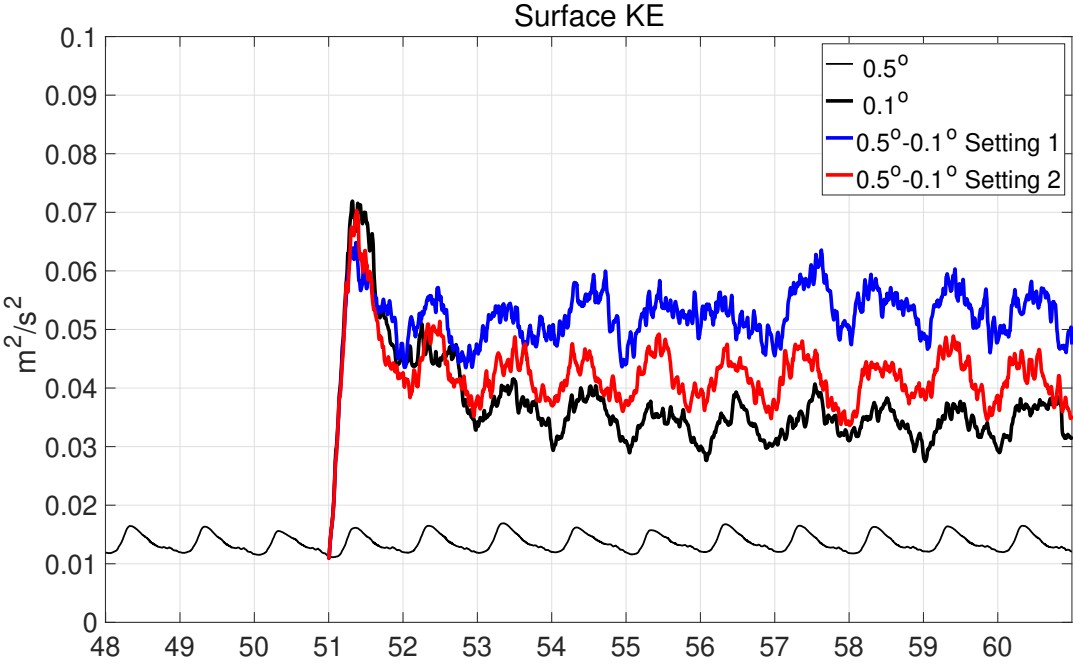

**Figure 6.** Mean surface kinetic energy of experiments with full-field $0.5°$ (**L**, thin black line), a spin-off full-field $0.1°$ (**M**, thick black line) since year 51, and two spin-off $0.5°$-$0.1°$ regional refinement experiments (**L-M-I** and **L-M-II**, in blue and red lines, respectively).




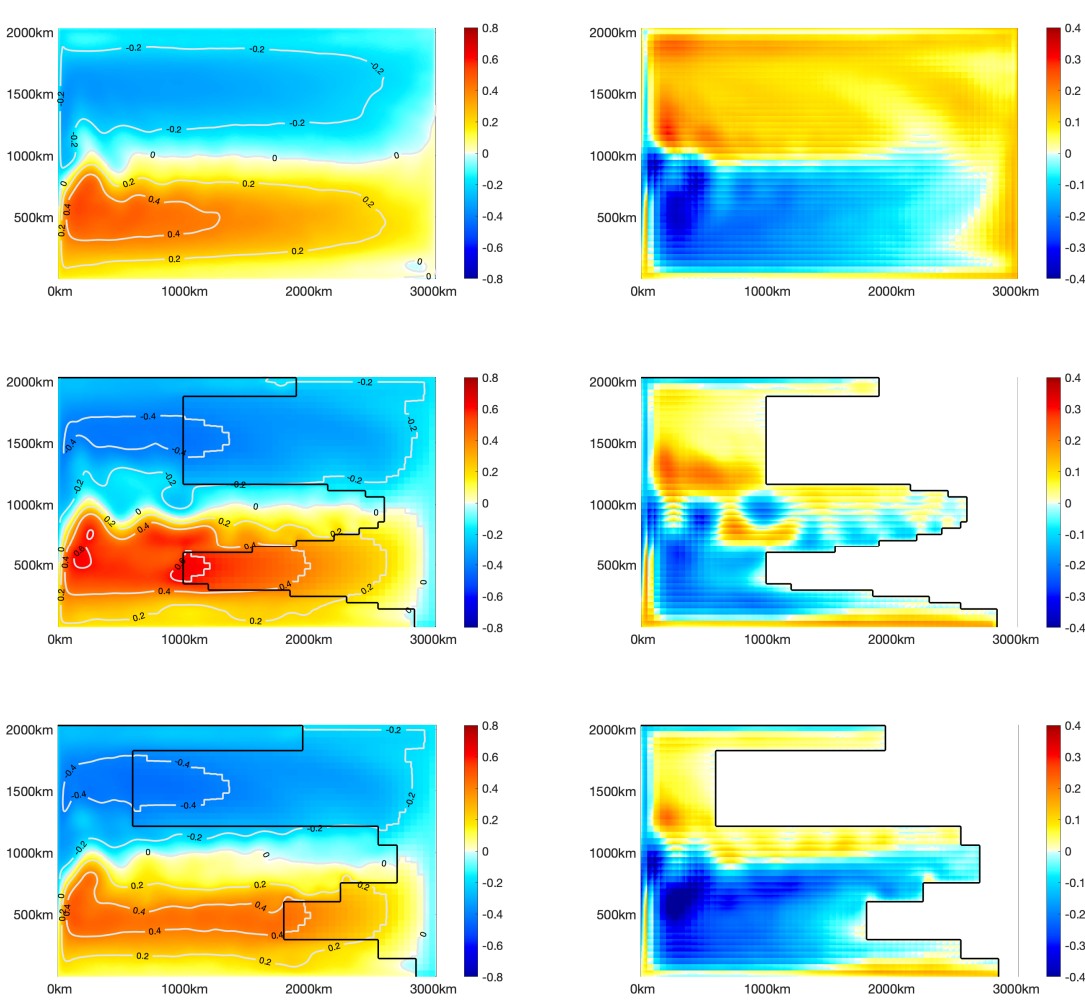

**Figure 7.** SSH climatology (left panels) of the full-field **M** experiment (top row) and two regionally refined 0.5°-0.1° experiments (**L-M-I** and **L-M-II** on the middle and bottom row). Difference with 0.5° simulations are shown on the right panels, respectively.



**Table 2.** Threshold values of surface velocity and surface relative vorticity (both in magnitude) for adaptive refinement from $0.1°$ to $0.02°$. Two settings are chosen, corresponding to the 80-th and 90-th percentile of the two parameters based on the full-field $0.1°$ experiment.

| Setting | Percentile | Velocity ($m/s$) | Relative Vorticity ($1/s$) |
|---|---|---|---|
| **M-S-I** | 80-th | 0.22 | $5 \times 10^{-6}$ |
| **M-S-II** | 90-th | 0.32 | $8 \times 10^{-6}$ |

It is worth noting that both the threshold values and the instantaneous fields are computed through spatial scaling. For example, in order to determine the threshold values, we accumulate 5-year's instantaneous fields of the $0.1°$ experiment, and compute the probability density of all the fields after spatial coarsening to $5 \times 5$ model grid cells (or $50km \times 50\ km$). The spatial scaling algorithms for the speed and relative vorticity is presented in Appendix B. Then, we choose 80-th and 90-th percentile for the absolute velocity and the absolute value of the relative vorticity as the threshold values (Tab. 2). The reason

why we use spatial coarsening is that the model's effective resolution is usually 5 to 10 times that of the grid's native resolution. By spatial coarsening of $5 \times 5$ grid cells on $0.1°$ fields, we attain a more physically realistic representation of both speed and vorticity at the scale of 50 $km$ (Rocha et al., 2016; Xu et al., 2021). Correspondingly, at each super cycle, we determine the instantaneous values of speed and vorticity at $50km$ scale and compare against threshold values for refinement.

Similar to the $0.5°$-$0.1°$ experiments, we carry out three spin-off experiments from $0.1°$ to $0.02°$ in both boreal winter and

summer. With full-field $0.1°$ experiment, on Feb.-1st and Aug.-1st, we carry out a full-field $0.02°$ experiment, as well as two adaptive refinement experiments, denoted **M-S-I** and **M-S-II**, respectively (Tab. 2). Figure 8 shows the mean surface KE of the three $0.02°$ refinement experiments, up to 3 months after refinement. After full-field refinement to $0.02°$, the model reaches the saturation of surface KE within two months. During winter (starting from Feb-1st), the model simulates continued KE increase even after two months (i.e., after April), which is due to the inherent KE seasonal cycle (the black line for $0.1°$ as a reference).

For the experiment during both winter and summer, the surface KE is about 60% higher than the original $0.1°$ experiment.

Contrast to $0.5°$-$0.1°$ experiments, the refinement to $0.02°$ does not cause overshooting of KE (colored lines in Fig. 8). Furthermore, both AMR experiments show lower, but also closer KE values to the full-field $0.02°$ experiment (i.e., **S**) throughout each of the 3-month simulations. For refinement with 80-th percentiles of surface vorticity and speed (i.e., **M-S-I**), the model attains over 90% of the KE of the experiment **S**. The portion of refinement for **M-S-I** is temporally changing, and is mostly

lower than 50%. For comparison, with 90-th percentiles and **M-S-II**, the region of refinement is on average 30% of the whole basin, while attaining over 80% of the total surface KE. The following part of the section examines in detail the modeled submesoscale scale processes at $0.02°$.

**Submesoscale spin-up**

During winter, the experiments with $0.02°$ show prominent submesoscale development and associated features which are

not captured by the $0.1°$ experiment. In Figure 9, we show the instantaneous surface Rossby number (or $Ro$, $\zeta/f$) fields after 5 days of refinement to $0.02°$. Since adaptive refinement based on the instantaneous field at 0:00, Feb.-1st, the model has carried out 5-day's time integration with the same refinement region. During these initial days after refinement, the mesoscale patterns



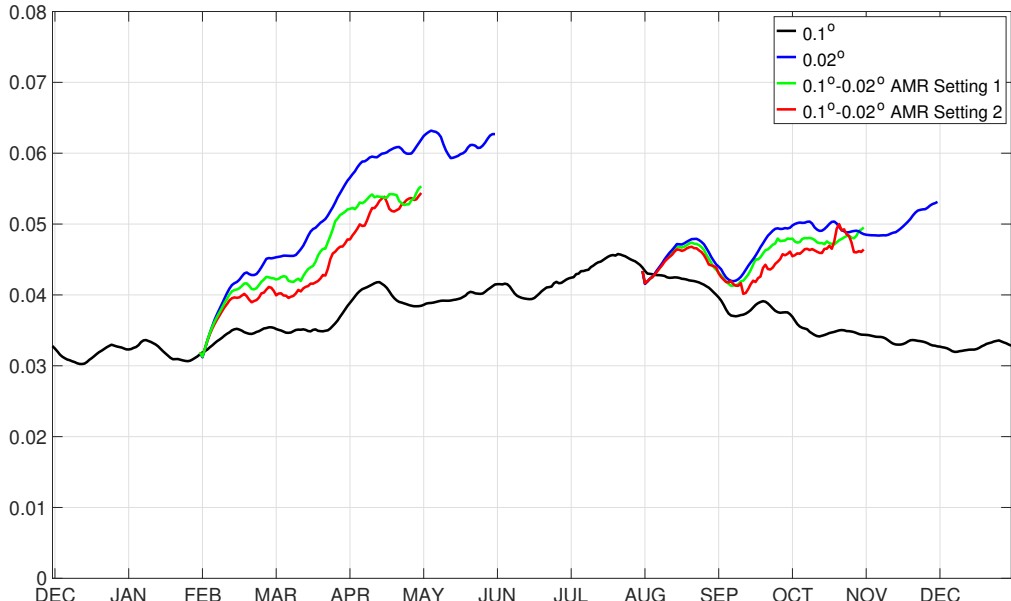

**Figure 8.** Surface mean kinetic energy ($m^2/s^2$) in $0.02°$ experiments and the associated $0.1°$ experiment.

across the three refinement experiments are very similar, and consistent with the full-field $0.1°$ experiment. The submesoscale processes are not fully developed in any of the $0.02°$-related experiments, with regions with large values of $Ro$ emerging from mesoscale features, such as fronts and other regions with large deformations.

For **M-S-II**, the region of refinement during winter mainly consists of the southern boundary and the WBC (Fig. 9.c). For the WBC, the region of refinement grasps the key features such as the mean flow, and the meandering WBC, as well as several strong mesoscale eddies. This indicate that with the threshold value-based criteria, the model is able to grasp the major region of focus, which is temporally changing mesoscale processes of WBC. The total area of refinement in Fig. 9.c is about 30%. Among all the grid cells, there are some cells that satisfy both refinement criteria for velocity and vorticity. This results in less than 20% of cells that are marked for refinement by the criteria. The higher refinement ratio at 30% during the experiment is caused by two factors. First, the lateral boundaries are marked for refinement to $0.02°$ by default (Sec. B). Second, there is automatic alignment of the refined region at $0.02°$ to 3 cells at $0.1°$ in each direction, in order to attain more regular patches at $0.02°$. Both factors contribute to the larger ratio of refinement of about 30%.

For **M-S-I** (panel b in Fig. 9), we use smaller threshold values, and as a result the region of refinement is larger and extends to other parts of the basin, including: (1) more portion along the southern boundary, (2) a larger portion on the WBC and its extension, and (3) other regions with intermittent high speed and/or vorticity, such as those in the north-eastern part of the basin. Especially, the refined region on the basin's southern boundary in **M-S-I** include more part of the strong flows that branch off the boundary and extend into the basin. Besides, the refined region on the WBC extends to the east and to either side of WBC,





and very close and even linked to the refined region on the basin's southern boundary. Overall, the region of refinement is 46% of the basin, which, in turn, is larger than the maximum refinement ratio by the threshold values at 40%.

Compared with full-field $0.02°$ experiment (panel a of Fig. 9), the major hot spots on the WBC and the boundaries are well captured bu the two AMR experiments. The larger region of refinement in **M-S-I** than in **M-S-II** at 5-day is indicated in marginally higher surface KE in the former experiment (Fig. 8). Also, both experiments have smaller, but close KE than the

full-field $0.02°$ experiment.

After another 15 days (or, 20 days since Feb-1st), the overall mesoscale pattern remains consistent across all four experiments (Fig. 10). At this time, the model has undergone 4 full super cycles of refinement (i.e., each cycle of 5 days). In AMR experiments, the major part of the WBC is always refined during the 20-day period, due to the constantly large magnitude of surface velocity and vorticity in this region (see also the corresponding video supplementaries). As a result, the submesoscale

processes in WBC undergo continued development, and they show the same features as the full-field $0.02°$ experiment. These features include the returning flow at $(400km, 800km)$ and the large cyclonic eddy at $(400km, 500km)$. In all three $0.02°$ experiments, all the ocean fronts and filaments along the WBC and major submesoscale are more sharpened compared with $0.1°$ counterpart. Certain areas with small-scale vorticity structures are modeled with full-field $0.02°$, but not captured by AMR experiment. Examples include the south-eastern part of the basin at $(2000km, 500km)$. This region has not always been marked

for refinement during the 20 days (therefore no continued spin-up), mainly due to the specific choice of threshold values. In terms of KE, the two AMR cases attains over 85% of the total KE attained by full-field $0.02°$ after 20 days of refinement (Fig. 8).

After 50 days of refinement, the submesoscale processes in $0.02°$ regions in **S**, **M-S-I** and **M-S-II** are well developed (Fig. 11). For full-field $0.02°$, a large band rich in submesoscale features is present, with a continuum of small-scale vortices in the

WBC extension and towards the east end of the basin. Certain part of this band, as well as the WBC is captured in **M-S-II** (i.e., 80-th percentiles). For comparison, in **M-S-I**, the major part of this band is not marked for refinement. It is worth to note that, the refinement criteria are based on the field as simulated at $0.1°$. During winter, the submesoscale processes and the ensuing kinetics are mainly driven by mixed layer instability (Khatri et al., 2021). At $0.1°$ (or 10 $km$) resolution, the model cannot directly simulate these processes. Potentially, the refinement criteria can be augmented for these processes, if they are

the subject of study with AMR.

At 50 days, we are approaching predictability limitation for ocean's mesoscale. Although the overall mesoscale pattern retains certain similarity across the experiments, noticeable differences emerge, especially between $0.02°$ and $0.1°$. For example, the location of WBC branching off the west coast differs: at $500km$ for $0.02°$ experiments, and at $700km$ for $0.1^circ$ experiment. We further compare and analyze a typical transect in these experiment in the following up part of the paper.

After 50 days, the difference between $0.02°$ and $0.1°$ is also more pronounced. Within the region of $0.02°$, the model simulates overall stronger flow, sharper and finer structures. Since we do not include feedback of $0.02°$ onto $0.1°$ regions, in AMR experiments, the region with $0.02°$ is actually forced by the full-field $0.1°$ experiment through $0.02°$-$0.1°$ boundaries. The difference in model states at 50 days results in inconsistencies along the boundaries, including artificial convergence and gradients. Examples include the southern boundary of the WBC in Figure 11. The 'noises' on the resolution boundaries are not





evident during the previous stages of refinement (i.e., Fig. 9 and Fig. 10, as well as video supplementaries), further indicating that they are caused by model state inconsistencies. We consider these noises are of numerical nature, and they can be reduced through interactions between the resolutions. We further discuss the future development of OMARE to support two-way and feedback of $0.02°$ on $0.1°$ in Section 4.

Submesoscale processes in the subtropics usually have pronounced seasonality. For comparison with winter, we show in
Figure 12 the surface relative vorticity after 50 days of refinement during summer (i.e., since Aug. 1st). Contrast to the winter, the surface KE in all refinement experiments are very close during summer (Fig. 8). The lower KE values correspond to the lower buoyancy input during summer, which greatly energizes the surface ocean during winter. Accordingly, the summertime vorticity field with $0.02°$ is much muted, with the absence of small-scale, ageostrophic structures (Fig. 12.a). Also the boundary-related numerical noises emerges as the AMR refinement experiments progress (see also the video supplementary for earlier
stages of refinement during summer).

**Analysis of a typical transect**

We further examine the mesoscale and submesoscale processes of the WBC in $0.02°$ experiments. We pick the a typical meridional-vertical transect for each experiments after 50 days of refinement to $0.02°$. The transect is about $550km$ from the west end of the basin for the full-field $0.02°$ experiment (Fig. 11.a). The transect traverses: (1) the southern boundary, (2)
the subtropical gyre which include large mesoscale features including a prominent coherent structure as an anticyclonic eddy (marked by **I**), (3) the eastward main axis of WBC (marked by **II**), (4) a kinematically active, submesoscale-rich part north to the WBC (marked by **III**), and (5) the subpolar gyre and the northern boundary of the basin. Due to the diverging WBC meandering among the experiments after 50 days, we adopt a small offset ($50km$) to the east for the other three experiments, in order to align these marked features to the all-field $0.02°$ experiment. As shown in Figure 11.b and c, the corresponding
mesoscale features are well captured with the transects with modified locations.

Figure 13 shows the temperature and velocity structure on the meridional-vertical transects in Figure 11 (surface 300 meters). The large anticyclonic eddy is well captured, although the meridional location differs among the experiments (thick red line in each panel). The eddy is more intensified to the surface, indicated by the higher zonal speed and also isothermal lines (e.g., $17°C$). For **M-S-I** (second panel), the cyclonic eddy is partially traversed by the transect, which is relatively weaker and to the
north of the large cyclonic eddy as in **S**. For all three $0.02°$ experiments (full-field and AMR), the eddy is evidently stronger than the $0.1°$ experiment.

The vertical speed ($w$) at $50m$ depth around this anticyclonic eddy indicates intensified submesoscale activities (Fig. 14). The large value as well as the spatial variability of $w$ around and within this eddy is mainly associated with the filaments and eddy boundary and ensuing fronts. Similar to the overall eddy intensity, the extreme value of $w$ is in the range between 40
$m/d$ and $60\ m/d$ for the $0.02°$-related experiments, which is more pronounced than the $0.1°$ experiment which has $w$ within $20\ m/d$.

Further to the north along the transect is the WBC core. The zonal locations of the WBC core are close in **S** and **M-S-I**, both at about $700\ km$ from the basin's southern boundary. For comparison, for **M** and **M-S-II**, the WBC core is to the south, at about $610\ km$ and $630\ km$ from the southern boundary. All four experiments show similar strength of the eastward flow of




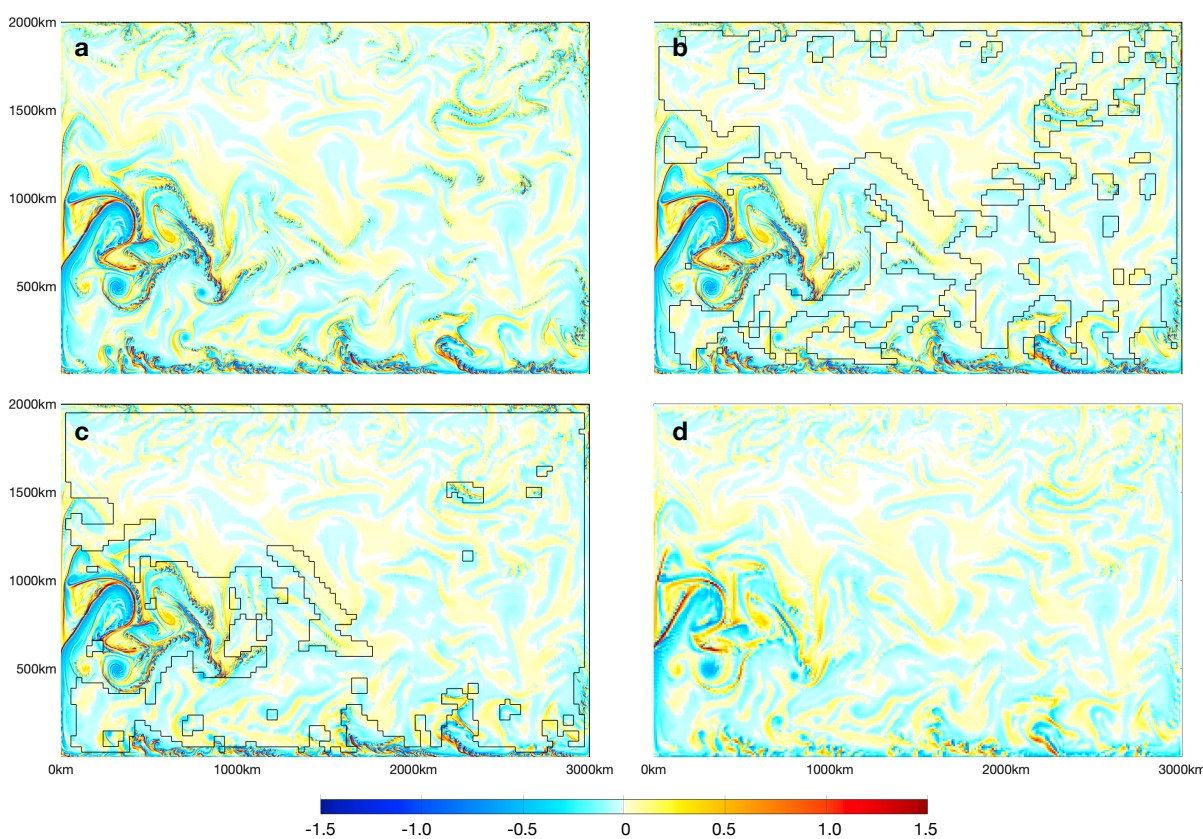

**Figure 9.** Surface Rossby number ($\zeta/f$) after 5 days of (adaptive) refinement from $0.1°$ to $0.02°$ since Feb.-1st. Panel a, b and c shows the result for full-field $0.02°$ experiment (i.e., **S**), that of AMR setting 1 (i.e., **M-S-I**), and that of AMR setting 2 (i.e., **M-S-II**), respectively. In both AMR experiments (panel b and c), the boundary between $0.02°$ and $0.1°$ is marked by black lines. Panel d shows the reference simulation result at $0.01°$ on the same day.



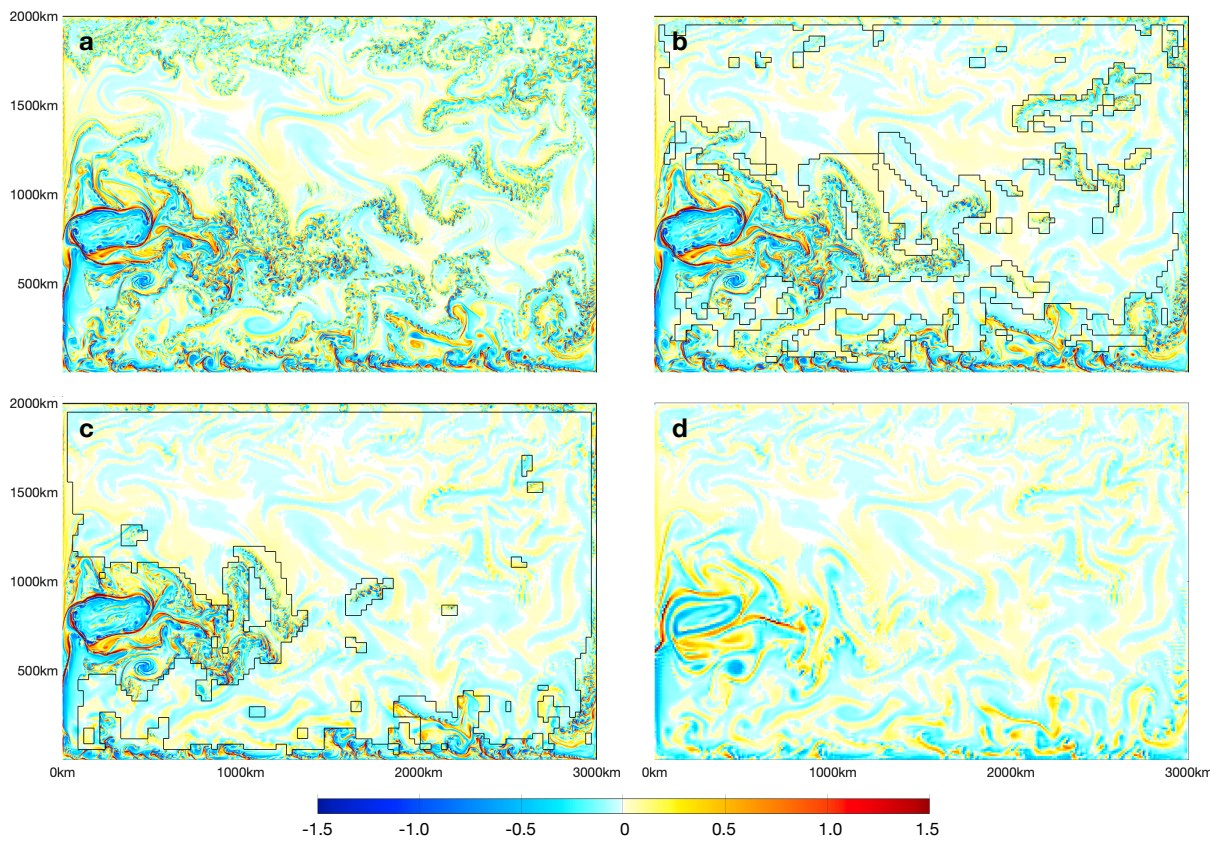

**Figure 10.** Same as Fig. 9, but for 20 days after the refinement since Feb-1st.

the WBC core of over 1.5 $m/s$ (Fig. 13). The zonal temperature gradient is also similar, with a sharp transition from $16.5°C$
to $12°C$ within $50\ km$ for all three $0.02°$ experiments, and about $100\ km$ for the $0.1°$ experiment.

The absolute value of vertical speed is over $100\ m/d$ for **S** and **M-S-I** on the WBC core (Fig. 14). For **M**, the largest vertical
speed on the whole transect manifests at the WBC core, at about $25\ m/d$. Compared with $0.02°$ experiments, the relatively
lower vertical speed in the $0.1°$ experiment is consistent with the weaker zonal temperature (as well as density) gradient across
the WBC core.

To the north of WBC core, there is a region with intensive submesoscale activities as modeled at $0.02°$, with small-scale
structures of temperature gradients and vertical motions. At mesoscale, this region (marked **III**, green bars in Fig. 13) consists
of an eastward flow flanked by two westward flow to its south and north. This structure is captured at $0.1°$, although the
submesoscale features are not present at this resolution. This is indicated by that both temperature gradient and vertical speed
show small-scale variability with large values in $0.02°$ experiments.




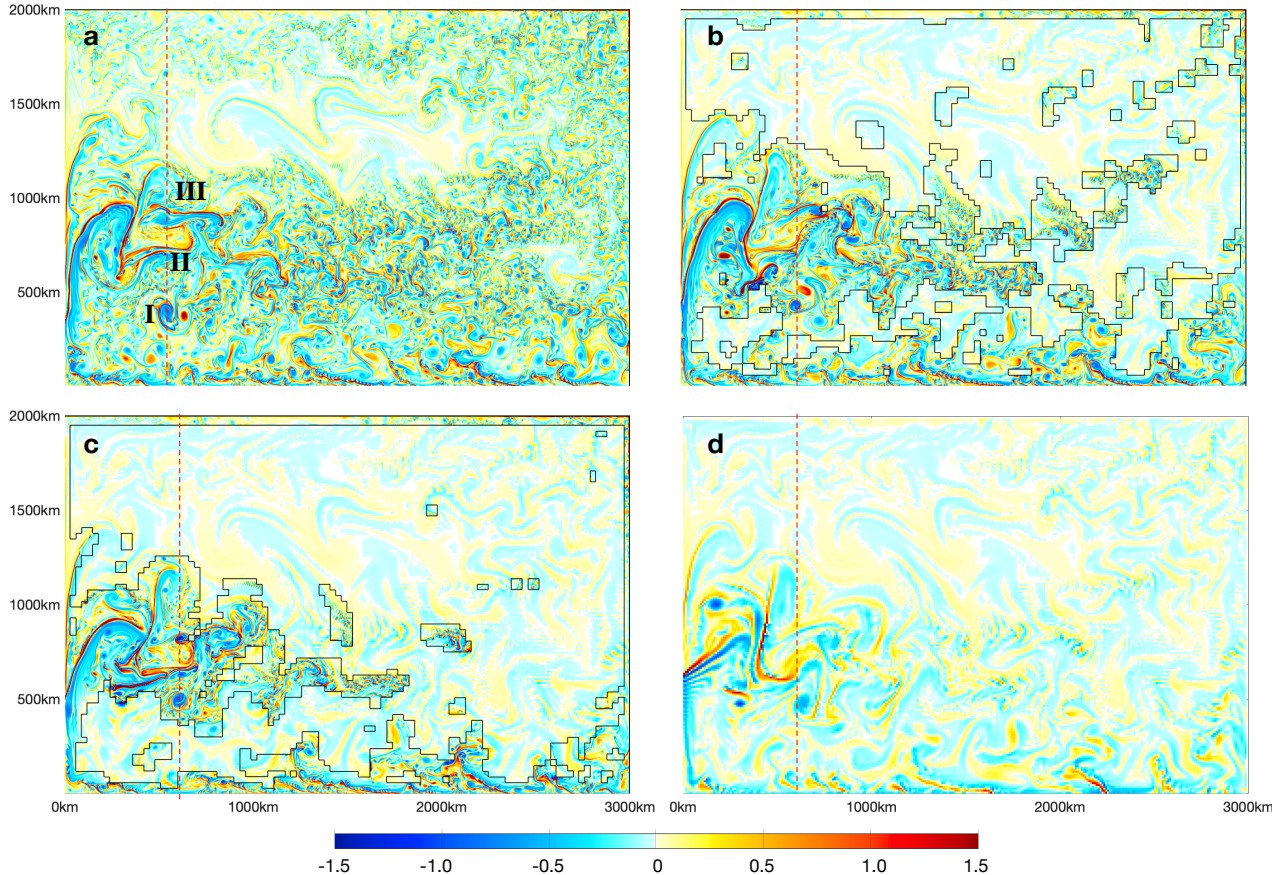

**Figure 11.** Same as Fig. 9, but for 50 days after the refinement since Feb-1st. In each panel, a meridional transect is marked by a dashed red line. In order to improve the consistency of mesoscale features along these transects, in panel a, the location of the transect is $550km$ from the western boundary, and in other panels, the location is offset to the east by $50km$, at $600km$.

One outstanding issue in **M-S-II** is the large zonal speed accompanied with large vertical speed ($>100$ $m/d$) at the zonal location of $800$ $km$. Despite the smaller refined region to $0.02°$ in **M-S-II** than **M-S-I** or **S**, as shown in Figure 14, the vertical speeds are higher in this region (marked by **III** in Fig. 11). This large vertical speed corresponds to a very strong small eddy that has been shed from the coarse/fine grid boundary nearby (about $50$ $km$ to the east in Fig. 11). Similar problems are not witnessed during earlier phase of refinement (i.e., Fig. 9 and 10). We conjecture that this is of numerical rather than physical origin, and potentially caused by: (1) the one-way coarse/fine forcing in the current version of OMARE; and (2) the specific refinement region in **M-S-II** at this stage. As mentioned above, we do not have feedback of fine-resolution region back to the coarse resolution simulation. The overall mesoscale pattern gradually deviates among the four experiments, including WBC meandering and eddy locations. After 50 days, the stronger flow and deviated mesoscale features at $0.02°$ encounter inconsistent conditions provided by $0.1°$ on the boundaries. This potentially causes artificial ocean fronts and/or convergence/-





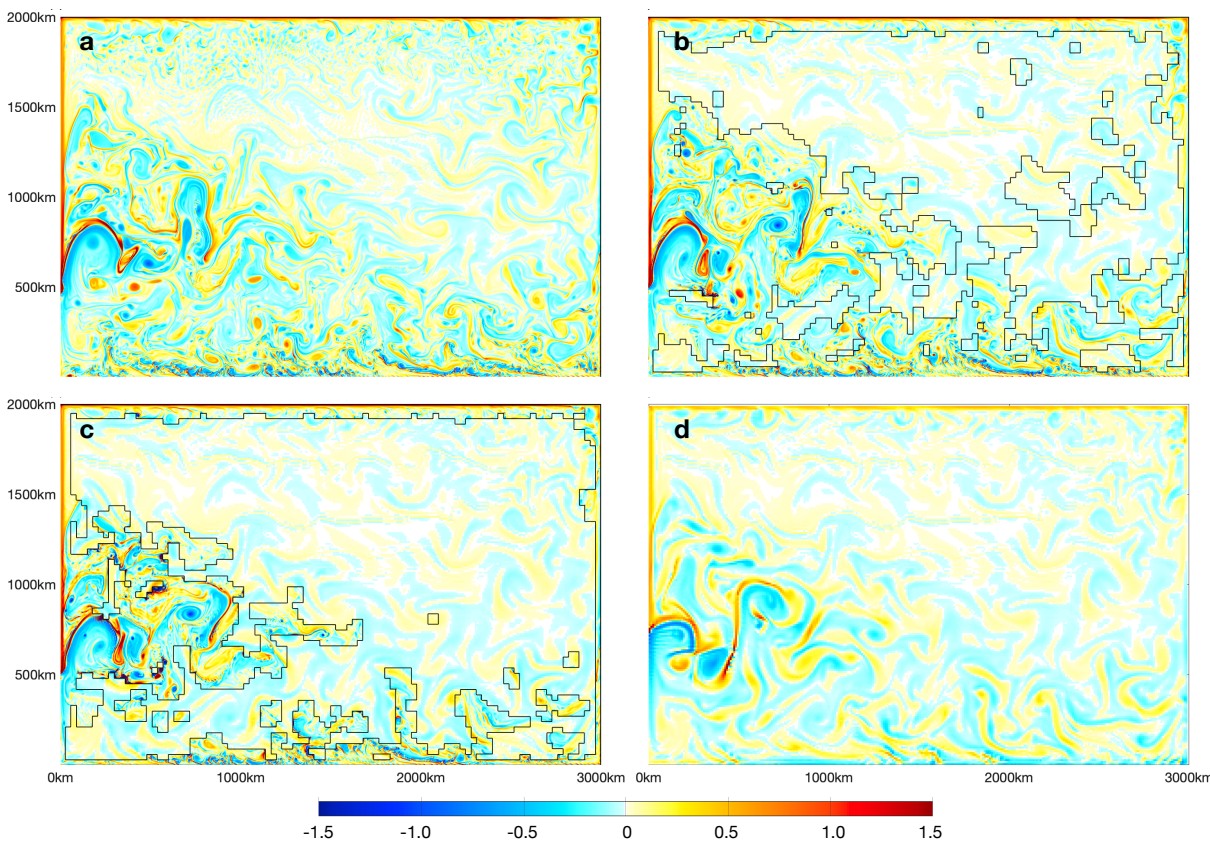

**Figure 12.** Same as Fig. 9, but for 50 days after the refinement since Aug-1st.

divergence, resulting in observed eddy shedding that are of numerical nature. Similar issues are witnessed during the later stages (i.e, 50-day's) of refinement during summer (Fig. 12). Further analysis of related experiments are planned in future studies. We also discuss in Section 4 the future work on improving the consistency across resolutions in AMR with coarsening and updating.

Further to the north along the meridional transects, we encounter the subpolar gyre and the northern boundary of the basin. Most of the this region is not refined to $0.02°$ in the two AMR experiments, due to smaller surface speed and vorticity. Consistently, in the full-field $0.02°$ experiment, only on the far northern part (beyond 1600 $km$) some submesoscale features manifest. The northern boundary of the basin in the two AMR experiments are refined to $0.02°$ by default, due to the coarsening to 5×5 cells during the determination of refinement regions (Appendix B).





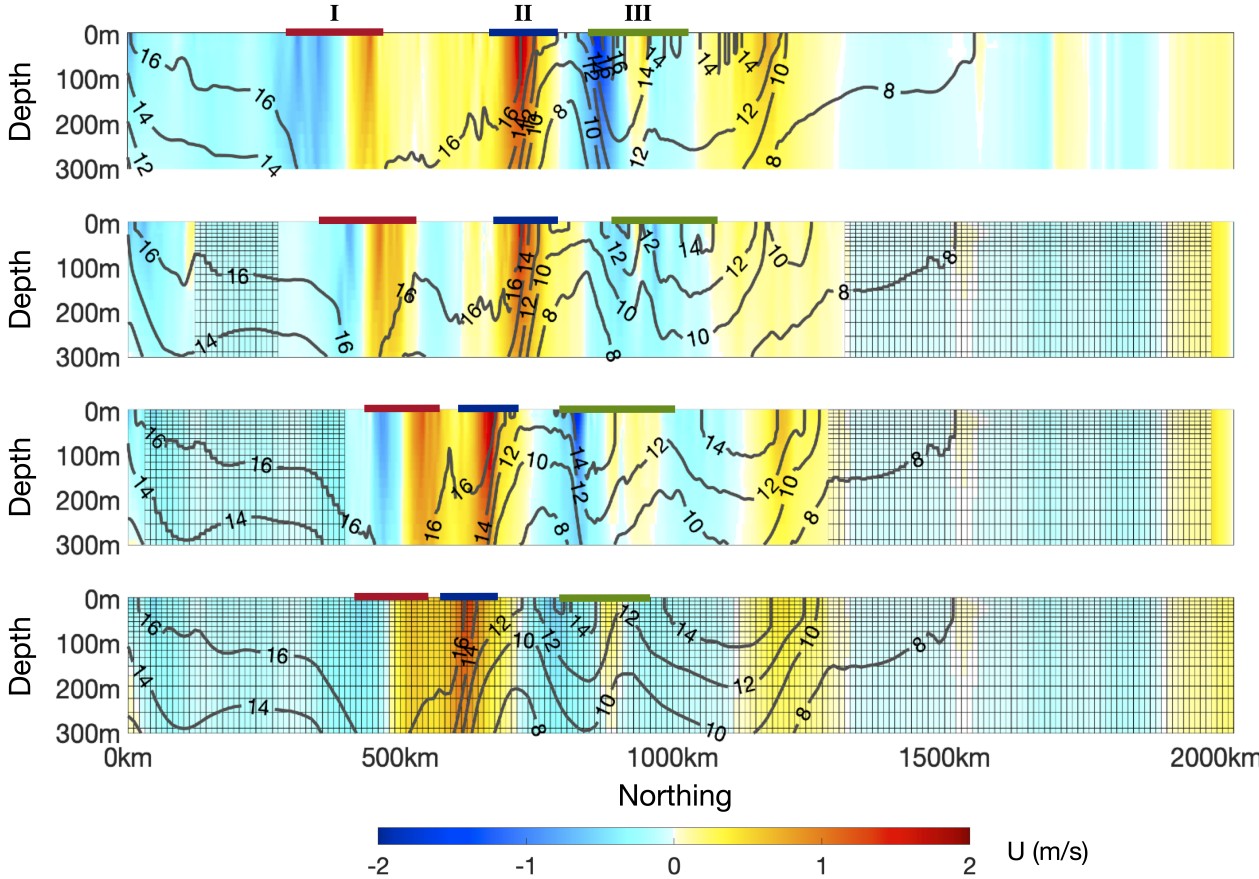

**Figure 13.** Typical meridional-vertical transects (top $300m$) in Fig. 11. Temperature (contour lines) and zonal velocity (filled contour) are shown in each panel. The four experiments, **S**, **M-S-I**, **M-S-I** and **M** are shown in order from top to bottom. Grids in the lower three panels show the region with $0.1°$ resolution (i.e., not refined). In each panel, the three features are marked: **I** for the mesoscale eddy in the subtropical gyre, **II** for the core of the western boundary current, and **III**) for the submesoscale-rich region to the north of **II**.

## 4  Summary and discussion

**Summary**

We present OMARE, an ocean modeling framework with adaptive spatial refinement based on NEMO, with the initial results based on an idealized double-gyre test case. Compared with AGRIF-based NEMO, we adopt a third-party software middleware of JASMIN to satisfy various modeling needs. JASMIN provides NEMO with the service of adaptive refinement, as well as an abstraction layer that shields away details of domain decomposition, parallel computing and model output. This paper mainly focuses on the porting of NEMO onto JASMIN, as well as the ocean physics at three resolutions of $0.5°$, $0.1°$ and $0.02°$, which are typical in climate studies and high-resolution simulations. We investigate in particular the (adaptive) mesh refinement with





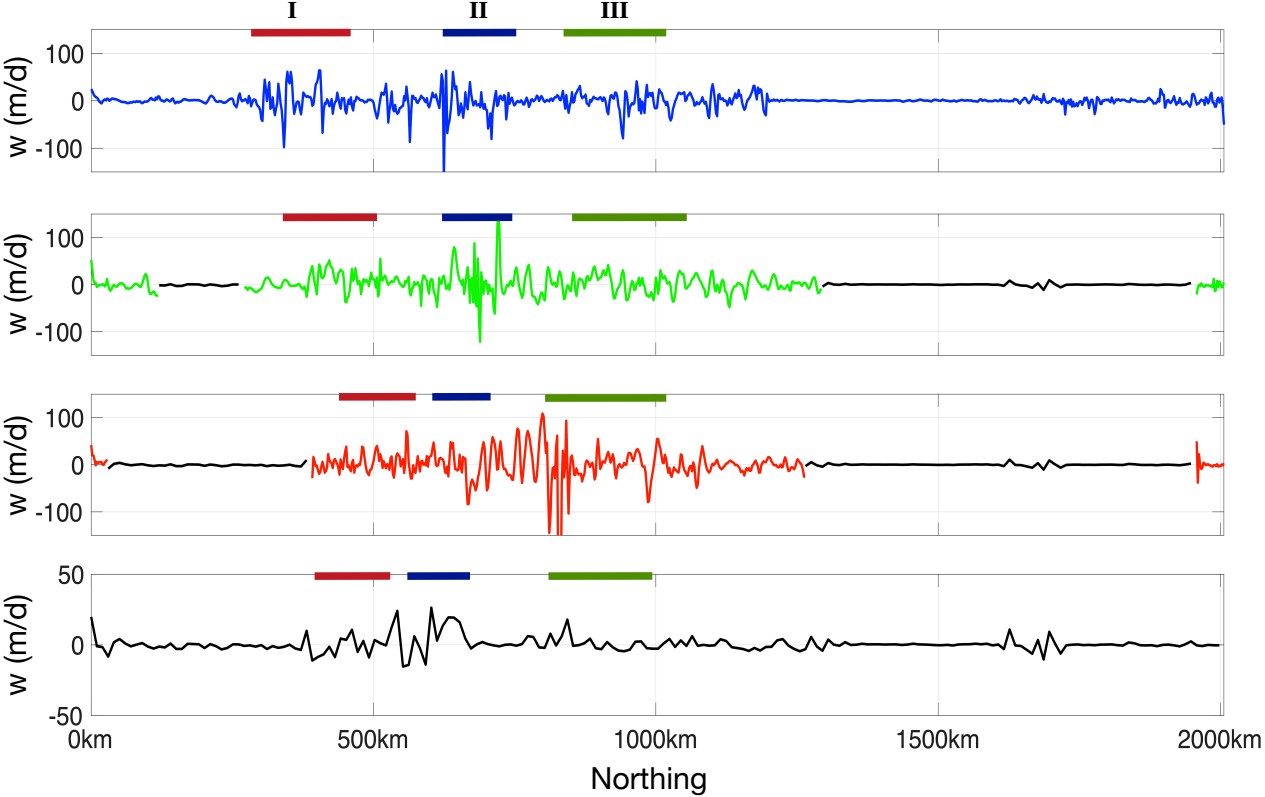

**Figure 14.** Vertical speed ($w$, $meter/day$) at $50m$-depth ($k$=7) on the meridional-vertical transects in Fig. 13. Panel layout, as well as the three features are the same as in Fig. 13. In the lower three panels, the region with $0.1°$ ($0.02°$) resolution is marked by black lines (colored lines). Note the range is different in the bottom panel, due to the overall smaller vertical speed in the full-field $0.1°$ experiment (i.e., $M$).

these resolutions in a double-gyre test case which simulates a western-boundary current system. A follow-up paper (part 2) will further analyze the computational aspects of OMARE.

The code refactorization onto JASMIN involves a non-trivial process of mixed language programming with FORTRAN and C++. The major work is divided into two parts: (1) to transform the NEMO into decomposition-free components that satisfy JASMIN's protocols, and (2) rewrite the time-integration in C++ with JASMIN. JASMIN components are elements for communication (i.e., boundary exchanges) and computation. We follow the JASMIN routines, and refactor both the dynamic core of NEMO and the various physical parameterization schemes that are needed for the 3 resolutions. In total the refactorization

yields 155 JASMIN components and 422 JASMIN patches, with over 30'000 lines of code (in both FORTRAN and C++). For comparison, for AGRIF-based NEMO, the time integration is also managed externally (i.e., by AGRIF), but the overall integration is similar to an 'add-on' to NEMO and relatively lighter in terms of coding compared with OMARE. Besides, the parallel input/output and restart is managed through JASMIN, instead of a standalone library of XIOS in NEMO (or AGRIF-based NEMO).



For the ocean physics as modeled with OMARE, with the refinement from 0.5° to 0.1°, the model simulates drastically different physical processes, from laminar to mesoscale-rich, turbulent ocean. Boundaries between the two resolutions act as a lateral forcing for the 0.1° region, providing boundary conditions for the fine-resolution region through both spatial and temporal interpolation. As a result, both the mean status in the subpolar and subtropical gyres and the kinematics at 0.1° are directly affected by the status simulated with 0.5°. The higher potential energy at lower resolution is also witnessed in Marques

et al. (2022) (Fig. 3 of the reference showing higher PE at 1/4° than higher resolutions). This is consistent with our study, indicating that: in low resolution models, the energy is removed by grid scale diffusion and not effectively converted to kinetic energy, hence the accumulation of PE.

We further demonstrate the adaptive refinement from 0.1° to 0.02°, focusing on the temporally changing processes on WBC. With the threshold value-based refinement criteria of velocity and relative vorticity magitude, the model grasps the major

mesoscale features on WBC. Furthermore, reasonable submesoscale processes are simulated by refining to 0.02°, and in particular, their seasonality. Outstanding issues include the inconsistencies across the resolution boundaries beyond the mesoscale predictability, as well as the ensuing numerical noises. These issues, among other related topics, are further addressed in the discussion below.

**Refining criteria and frequency for AMR**

The refinement criterion for AMR is an open question for simulating the ocean's multi-scale processes. In Section 3.4 we showed that the instantaneous surface velocity and vorticity at 0.1° serve as good proxies for mesoscale processes on WBC. Further improvements to the criteria can be achieved by introducing memory and predictive capabilities to better capture the mesoscale features. Besides, the threshold values can be further improved, instead of using percentiles in Section 3.4, e.g., a prescribed KE amount/percentage with AMR. Instead of AMR, a prescribed refinement region based on a priori knowledge

is another type of criteria (Sec. 3.3). OMARE support both types of refinement (dynamic or static), through a generalized interface for marking arbitrary grid cells for refinement. Due to the inherent turbulence of the ocean's mesoscale, the specific regions of interest, such as mesoscale eddies, WBC meandering, and ocean fronts, are temporally changing and have process-dependent predictability. Therefore, the support for AMR in OMARE ensures the flexibility in future application in various modeling scenarios and processes of even finer temporal/spatial scales.

The update frequency of refinement region is another key parameter for AMR. For the refinement from 0.1° to 0.02°, we use five days as the update interval (i.e., super cycle). This interval should be no larger than the temporal scale of the process to be studied, here, the mesoscale processes. In general, with longer intervals, the refinement region should also be enlarged, in order to accommodate the changes in the locations for the regions of interest. Another contributing factor is the computational overhead associated with adjusting the refinement region. For OMARE and JASMIN, this overhead include

the domain decomposition and (re)mapping to processors, the establishment of communication framework, and the associated model state migration and interpolations. We will further examine the computational behavior of various AMR scenarios in part 2 following this paper.

**Boundary exchanges**



For the lateral boundaries between coarse and fine resolutions, we have implemented bilinear interpolators for both state
variables on the outer boundaries (i.e., $T$, sea-surface height, among others) of the fine-resolution grid and fluxes on the
boundaries (i.e., $u$ and $v$). For temporal interpolation, a simple linear interpolation is utilized, with support for both barotropic
and baroclinic steps. Better, more sophisticated schemes are available from the established works involving NEMO and AGRIF.
In particular, Debreu et al. (2012) has meticulously designed conservative interpolators, filters and sponge layer, and specific
treatments to the split-explicit formulation of the dynamic core. In futur work, we plan to investigate and implement improved
interpolators, and further evaluate their numerical behavior in OMARE.

**Upscaling in refinement settings**

Another future improvement to OMARE is the feedback from refined resolution to the coarse resolution. In OMARE, in
the region covered by fine resolution grid, the model still carries out simulation with the coarse resolution. A simple updating
scheme can be implemented, as follows: (1) coarsening with the model status in the refined region, and (2) overwriting the
model status on the coarse resolution. The updating process can lower the inconsistency of the two resolutions on the bound-
aries, hence reducing the noises as observed in Section 3.4. Besides, small-scale processes are unresolved and parameterized
on the coarse resolution. Therefore, from the physics perspective, the update also corresponds to an 'upscaling' process. The
refinement to a finer resolution yields more trustworthy model status due to explicitly resolving these processes. The updating
then potentially improves the model status on the coarse resolution, hence the effect of upscaling. Moreover, instead of direct
overwriting the model status on the coarse resolution, we can also adopt a data-assimilation assisted methods, such as nudging
the coarse-grid model with the fine-grid model status. Due to the inherently different ocean dynamics and even mean states
as simulated by different resolutions [e.g., Sec. 3.3, Levy et al. (2010)], how to improve the coarse resolution model with
refinement remains an open and daunting task. With AMR in representative regions and upscaling capability of OMARE, we
plan to carry out studies of key processes, including the effect of the restratification by submesoscale processes on the mean
status (Levy et al., 2010; Pennelly and Myers, 2020).

**Ocean physics in Double-Gyre and related idealized cases**

Double Gyre case is typical of wind-driven ocean circulation in the mid-latitude. There are several aspects of the case and
the corresponding model configurations that need further improvements. Unlike the two WBC systems in the Earth's northern
hemisphere, the Kuroshio and the Gulf Stream, the Double-Gyre is limited in size especially in the meridional direction. This
compromises the comparability to the realistic WBC systems, since we have witnessed prominent boundary-related mesoscale
and associated submesoscale processes for Double-Gyre case in this study(Sec. 3.4). Furthermore, the lateral boundary condi-
tion is uniformly free-slip in all experiments, which in effect inhibits the energy dissipation through lateral friction. However,
the *best* lateral boundary condition for ocean models, and specifically, for NEMO, remains elusive, and therefore subjected
to tuning to specific model settings, as well as targeted observations. Given the complex bathymetry for continental shelf in
realistic cases, there may be no general rule for the 'best' lateral boundary condition. OMARE, with the grid refinement on
lateral boundaries and upscaling, can be utilized to study eddy/current-bathymetry interactions, as well as the effective lateral
boundary condition.



Another aspects that needs improvements in OMARE is the specific choices of parameterization schemes and parameters. Lateral mixing scheme, such as GM90 (Gent and McWilliams, 1990) are widely used in non-eddying (i.e., laminar) ocean

models in order to approximate the effect of mesoscale eddies and improve the model's simulation of isopycnal mixing. In OMARE, we currently use a simple, 1st-order Laplacian mixing scheme for the sake of simplicity. At higher resolutions (0.1° and 0.02°) we adopt the second-order mixing scheme with adaptation to grid cell sizes. The parameters are chosen by apriori experience, and should be scrutinized for further tuning of the model. This also applies to the TKE scheme for vertical mixing which is used across all three resolutions.

Double-Gyre as used in this study belongs to a series of idealized test cases for ocean models, such as Levy et al. (2010) and Marques et al. (2022). Especially, the rotation of the gyre to be the purely zonal-meridional direction from Levy et al. (2010) enables a more regular-shaped double-gyre system, but disables the polar branch of the system. Also (Marques et al., 2022) proposes an idealized case of intermediate complexity, which contains northern and southern hemisphere, the tropical ocean, as well as the circumpolar circulation with an east-west periodic boundary condition. However, the atmospheric forcing in

Marques et al. (2022) does not contain a seasonal cycle. For comparison, the seasonal cycle of the forcing in this study enables the simulation of the seasonality of submesoscale activities.

**Realistic cases**

In order to carry out simulation of earth's ocean with OMARE, realistic bathymetry, ocean states, as well as historical atmospheric forcings should be incorporated in OMARE. Besides, a prognostic, fully thermodynamic and dynamic sea ice

component is also needed, such as SI$^3$ (SI3) which is a module in Surface Boundary Condition (SBC) of NEMO. We plan to incorporate sea ice in the future version of OMARE, as well as other relevant processes including tide and wind wave. Both static and adaptive refinement can be further utilized for the study of the key regions and/or multi-scale processes of the global ocean.

*Acknowledgements.* The authors would like to thank the editors and referees for their invaluable efforts in improving the manuscript. This

work is partially supported by: National Key R & D Program of China (No.: 2017YFA0603903 and No.: 2017YFA0603902), the Program of Natural Science Foundation of China (no.: 42030602). The authors would also like to thank the National Supercomputing Center at Wuxi for the computational and technical supports for the numerical experiments.

*Code availability.* OMARE is based on the NEMO source code (version 4.0.1) which is available for download at: http://forge.ipsl.jussieu.fr/nemo/browser/NEMO/releases/release-4.0.1 (last accessed: May 3, 2022). JASMIN is a closed source software, which can be applied for

usage at: http://www.caep-scns.ac.cn/JASMIN.php (last accessed: May 2, 2022). The codebase of OMARE, along with the output of three experiment (in HDF5 format), are available on Zenodo at https://zenodo.org/record/6699768 (doi:10.5281/zenodo.6699768, last accessed: Jun 22, 2022). The code of OMARE is publicly available under GPLv3 licence (https://www.gnu.org/licenses/gpl-3.0.txt).



$$\tau_x(\phi,t) = 8.7 \cdot 10^{-4} ((\cos(2\pi t - 0.79) + 12))^2 \cdot \sin(0.38\phi - 6.82 + 0.5\cos(2\pi t - 0.79)) \tag{A1}$$

$$SST_f(\phi,t) = 28.3 \cdot \frac{(1 + 0.02 \cdot \cos(2\pi(t - 0.558))) \cdot \cos(\pi \cdot \phi - 5)}{107 + 22.5 \cdot \cos(2\pi(t - 0.558))} \tag{A2}$$

$$Q_{tot}(\phi,t) = -40 \cdot (SST_f(\phi,t) - SST(\phi,t)) = Q_{sr}(\phi,t) + Q_{ns}(\phi,t) \tag{A3}$$

$$Q_{sr}(\phi,t) = 230 \cdot \cos(0.019 \cdot \phi - 0.447 \cdot \cos(2\pi(t - 0.475))) \tag{A4}$$

*Video supplement.*  The daily instantaneous surface vorticity and surface height by spatial refinement from $0.5°$ to $0.1°$ are provided for the
3 experiments (**M**, **L-M-I** and **L-M-II**). The daily instantaneous animation of adaptive refinement experiments from **M** to **S** are provided for
both summer and winter (i.e., **M**, **S**, **M-S-I** and **M-S-II**).

*Author contributions.*  SX conceived and designed the overall framework of OMARE. SX, DT and HA carried out the model design. YZ,
YS, SX, DT, HG, and HA carried out model code refactorization, porting and testing. YZ, YS and SX carried out the numerical experiments
and analyzed the results. All the authors contributed to the writing of the manuscript.

*Competing interests.*  The authors declare that they have no conflict of interest.

**Appendix A:  Atmospheric forcings for Double-Gyre case**

The atmospheric forcing of the double-gyre case contains the wind stress forcing and the thermodynamic forcing which both
contain an annual cycle. Each model year contains 360 days. The wind stress is purely zonal (i.e., $\tau_x$, in $N/m^2$) and a function
of both latitude ($\phi$, in radians) and time of the year ($t$, normalized between 0 and 1 within the year), as shown in Eqs. A1. The
meridional wind stress is constantly 0 $N/m^2$. The thermodynamic forcing also follows a meridional structure, and enforced
on the model's surface through a restoring condition to the predefined annual cycle of apparent temperature, $SST_f$, defined in
Eqs. A2. The restoring strength is $-40 W m^{-2} K^{-1}$. Total surface heat flux (i.e. $Q_{tot}$, split into solar part $Q_{sr}$ and non-solar
part $Q_{ns}$, both in $W m^{-2}$) is in proportional to $SST_f - SST$ with $SST$ the model's instantaneous surface temperature. $Q_{tot}$
is defined in Eqs. A3, following Levy et al. (2010). Solar heat flux $Q_{sr}$ in Eqs. A4 penetrates into sea water, while non-solar
heat flux $Q_{ns}$ only influences the surface layer of the model. The most extreme conditions of $\tau_x$ and $SST_f$ during winter and
summer are shown in Fig. 4.

In order to maintain the overall hydrological balance within the basin, we compensate the imbalance in fresh water with the
basin-mean value of evaporation-minus-precipitation (EMP). In the case of refinement experiments (including AMR), we only
compute the areal-mean value of EMP on the coarsest resolution, and use it as the freshwater forcing across the all resolution
levels.




$$\tilde{\zeta}|_{i_1,i_2;j_1,j_2} = \frac{\sum_{i=i_1}^{i_2} u_{i,j_1-1}dx_{i,j_1-1} - \sum_{i=i_1}^{i_2} u_{i,j_2}dx_{i,j_2} + \sum_{j=j_1}^{j_2} v_{i_2,j}dy_{i_2,j} - \sum_{j=j_1}^{j_2} v_{i_1-1,j}dy_{i_1-1,j}}{\sum_{i=i_1}^{i_2}\sum_{j=j_1}^{j_2} dx_{i,j} \cdot dy_{i,j}} \tag{B3}$$

### Appendix B: Spatial scaling for the computation of vorticity and velocity

We show the computing of relative vorticity related spatial derivatives with the two-dimensional velocity fields and Arakawa-C grid staggering. The relative vorticity ($\zeta$) at a certain spatial scale is defined as the combination of two derivatives at the specific scale: $\zeta = -\frac{\partial u}{\partial y} + \frac{\partial v}{\partial x}$. The two derivatives are formally defined by line integrals over a certain area $A$ with the corresponding spatial scale of $\sqrt{A}$, as in Eqs. B1 and B2.

For NEMO with Arakawa-C grid staggering in the horizontal direction, the relative vorticity is computed on the $f$-point (or North-East corner) of each cell. Fig. B1 shows the case of spatial scaling with 9 grid cells (i.e., 3×3) at the grid location $(i,j)$. Velocities of adjacent 16 cells are utilized to compute the two scaled derivatives of $\frac{\partial u}{\partial y}$ and $\frac{\partial v}{\partial x}$.

For the general case of computing the scaled version of relative vorticity (denoted $\tilde{\zeta}$) for the grid points from $i = i_1$ to $i_2$ in the $x$-direction and from $j = j_1$ to $j_2$ in the $y$-direction, we compute it following Eqs. B3. In Eqs. B3, the cell lengths of the $f$-points are denoted $dx_{i,j}$ and $dy_{i,j}$ in the $x$-direction and $y$-direction, respectively. The corresponding spatial scale is then computed as the square root of the total area, i.e, $\sqrt{\sum_{i,j}(dx_{i,j} \cdot dy_{i,j})}$.

For determining the refinement region for AMR, we compute for every cell in the grid the scaled values of the relative vorticity and the velocity, at the scale of 5×5 grid cells. These values are compared against prescribed threshold values. With spatial scaling, the line integral should traverse valid model cells. In the case of the area $A$ covering invalid cells (i.e., land), Eqs. B3 should be adjusted accordingly. For the sake of simplicity, we by default mark the cells that are adjacent to the lateral boundaries for refinement in this study.

$$\frac{\partial u}{\partial y}\Big|_{scale=\sqrt{A}} = -\frac{1}{A}\oint u \cdot dx \tag{B1}$$

$$\frac{\partial v}{\partial x}\Big|_{scale=\sqrt{A}} = \frac{1}{A}\oint v \cdot dy \tag{B2}$$



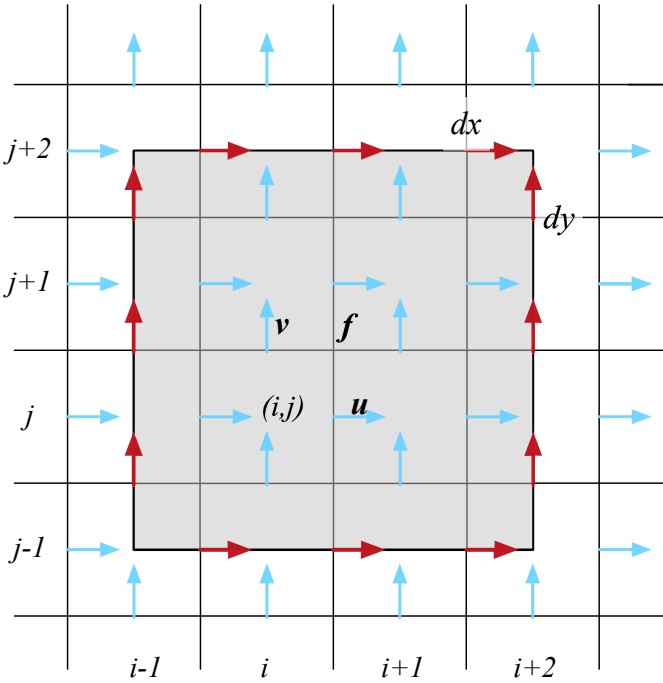

**Figure B1.** Spatial scaling of 3×3 cells with Arakawa-C grid staggering. All velocities used during the computation of line integrals are marked by red arrows, with others marked by blue.

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
