# Peer review of "Ocean Modeling with Adaptive REsolution (OMARE, version 1.0) – Refactoring NEMO model (version 4.0.1) with the parallel computing framework of JASMIN. Part 1: adaptive grid refinement in an idealized double-gyre case"

_EGUsphere, 2022_

## Author Comment (AC1)

**Author reply to the CEC1 to 'egusphere-2022-510' (Zhang et al., 2022)**

Dear Professor Juan Añel and Dr. Farneti,

Thank you very much for the comment. On behalf of all authors, I have the following reply and clarifications regarding the issues in your comment, mainly concerning the JASMIN software. The detailed reply is as follows, and in general we want to emphasize that, JASMIN is a third-party, infrastructural software of the model we presented (instead of the model itself), and we have updated the approaches of applying/using JASMIN, as follows.

(1) About the role of JASMIN as an infrastructure to our modeling study:

In the manuscript we introduced the model of OMARE, including the detailed process of constructing it, as well as initial results of numerical experiments. JASMIN is the environment that we construct and run the model. The scenario here, in our opinion, is similar to the that for porting a model (e.g., NEMO) onto a specific computational platform (e.g., CUDA). This example, very similar to ours, also involves the refactorization of model code, and code compilation/linkage as well as execution on the platform. Similar to JASMIN, the CUDA toolkit (available from NVIDIA, but closed-source, commercial software) serves as the infrastructure to the model development. Also similar to CUDA-alike software platforms, JASMIN is general-purposed, not specifically made for geophysical fluid models.

Furthermore, similar works to the platform-specific model development are actually very common in the GMD journal, which are all based on closed-source software. Not to mention the compilers and MPI environment, many of which are closed source. For the clarification from our point of view, given that the software (here JASMIN) serves as the model's infrastructure, its open-source status should not be a limiting factor, and it does NOT constitute compliance issues with GMD policy. Of course, the prerequisite is that the infrastructural software IS INDEED available for use/validation, which is addressed further below.

(2) On the open-source status of our work:

Upon the reminder from the editor, we immediately open-sourced the code, together with some of the output for public access. The newly added code (based on NEMO codebase) consists of about 100'000 new lines, in either FORTRAN or C++, and the whole source is already provided. Besides, due to the sheer volume of the model output, we could only publish some results, and the full set is readily available upon request.

We would like to emphasize that: we totally agree with the spirit of open-source for both contribution to the community and reproducibility/validation by peer scientists. At the same time, we are also aware of the various practical limitations of many existing papers (including recent ones), such as the licensing of model software to a limited (type of) audience, which are not uncommon on GMD or journals with similar focus. As model developers, we benefit from this movement, and we are definitely willing to contribute our efforts at GMD to the community.

(3) On the status & availability of JASMIN:

JASMIN is developed by Institute of Applied Physics and Computational Mathematics (IAPCM) and IACPM holds the full copyright to it. We, as external users of JASMIN, do not have access to the code, not to mention the right the disclose the code. Some of the authors of our paper are affiliated with IAPCM who helped with the software porting process, but this is an independent issue to the open-source status of JASMIN.

Currently JASMIN is distributed in the binary format. Third-party application, such as ours, are developed according to JASMIN's software API, and linked with the binary library of JASMIN. This is also common practice for using third-party software, and it is counterpart to the practice with open-source software which requires the end user to compile from scratch.

In order to overcome the availability issues of JASMIN as raised in CEC, we have provided the following two ways for applying JASMIN. First, one can apply for JASMIN by writing to: yang_zhang@iapcm.qc.cn. The JASMIN software will be provided through replies. Any inquiries/questions about using JASMIN and OMARE are also handled through this contact.

Second, since the English version of the webpage for JASMIN application is under construction, we provide the step-by-step instruction in the following pages. This guide is based on the original JASMIN application page at: http://www.caep-scns.ac.cn/JASMIN.php. And it provided in case that anyone interested in selecting specific version/platform for JASMIN.

We would like to express our sincere thanks again to the editors for their help. And we hope that our reply has fully addressed the comments by the chief editor.

Shiming Xu, on behalf of all authors

**An instruction of the webpage for JASMIN application**

**Step 0: Registration & Login**

中物院高性能数值模拟软件中心
CAEP Software Center for High Performance Numerical Simulation

首 页  中心概况  软件产品  人才招聘  品牌会议  联系我们

登录 | 注册
**Log in | Register**

**软件产品**

高性能科学与工程计算中间件 ∨
↳ JASMIN
↳ JAUMIN
↳ JCOGIN
↳ SuperMesh
↳ TeraVAP
↳ HiPro
↳ HiPESL

反应堆粒子输运软件系统 ＞
复杂电磁环境数值模拟软件系统 ＞
重大装备工程力学数值模拟软件系统 ＞
核相关/含能材料模拟与设计软件系统 ＞

产品介绍  成果展示  软件下载  升级更新  参考资料

**JASMIN**

**Click to log in or register**

**简要介绍：**

  JASMIN是面向结构网格应用研制的编程框架。该框架面向现代高性能计算机体系结构，设计数据结构、发展和集成高效并行算法、采用先进软件技术，提供屏蔽并行实现的编程接口，支持领域专家在个人电脑上"并行思考、串行编程"，快速研制并行应用软件。

  JASMIN框架已成功应用于武器物理、激光聚变、高新技术装备等领域，支撑了4个亿亿次和30多个千万亿次应用软件的快速研发和数值模拟。

物理模型 计算参数 计算方法 高效算法 ... 专家经验
物理数字个性层：实现JASMIN的抽象接口函数

支撑JASMIN实际应用的抽象接口函数
Interface Functions for User Applications
应用接口层

多物理时间积分算法 MultiPhysics Integrator
面向应用的计算工具箱 Application Utilities
解法器 Solvers
数值共性层

时间积分算法 Time Integrator
网格几何 Grid Geometry
数字运算 Math. Ops.

网格自适应 Mesh Adaptivity
数据传输 Communication
负载平衡 Load Balancing
并行自适应支撑层

网格片层次结构 Patch Hierarchy
数据片 Patch Data

基本工具箱 Toolbox

J A S M I N 框架

**功能特色：**

1. 支持多种网格类型：单块结构网格、多块结构网格（协调拼接、非协调拼接）、自适应加密结构网格；
2. 支持多种计算方法：h-自适应、r-自适应、接触碰撞、输运扫描、粒子模拟；
3. 集成多种解法器：线性、非线性、特殊解法器（FMM、FFT、Kohn-Sham、Poisson）及预条件库；
4. 支撑多物理场、多时空尺度耦合计算；
5. 支持异构协同计算；
6. 具备自动容错功能，支持"一次提交，完整模拟"；
7. 配备图形化集成开发环境，屏蔽并行编程和面向对象编程；

**应用实例：**

| 典型应用软件 | 应用领域 | 计算规模 | 处理器核数 |
| --- | --- | --- | --- |
| 时域全波电磁模拟软件 | 电磁环境 | 4608 亿网格 | 93 万异构核（12 万 CPU 核 +81 万 MIC 核） |
| 激光等离子体粗互作用粒子模拟 | 激光聚变 | 1555 亿粒子 | 93 万异构核（12 万 CPU 核 +81 万 MIC 核） |
| 激光等离子体粗互作用流体力学 | 激光聚变 | 805 亿网格 | 93 万异构核（12 万 CPU 核 +81 万 MIC 核） |
| 三维辐射流体力学界面不稳定性 | 激光聚变 | 70 亿网格 | 80 万异构核（10 万 CPU 核 +70 万 MIC 核） |
| 二维辐射流体耦合中子输运程序 | 武器物理 | 上亿自由度 | 1.2 万 CPU 核 |
| 关联电子体系模拟软件 | 材料科学 | 60 个 K 点 × 600 个特征态 | 1.2 万 CPU 核 |
| 三维冲击动力学粒子模拟 | 材料科学 | 2.56 亿粒子 | 3.2 万 CPU 核 |

**更新说明：**

JASMIN_LITE 4.0.0版更新内容：

1. 并行扩展性的提升，可有效实现20万核CPU上的大规模计算；
2. 将解法器分离出来，独立发布。

JASMIN 4.0.0版更新内容：

1.功能性能：新增了多层自适应和多块网格支持，并支持了FFT特殊解法器。除了一般解法器被移到JPSOL和不支持BMR之外，支持了JASMIN 3.2 版本的全部功能。
2.使用方式：匹配软件产品使用方式的标准化，对通过CMake使用JASMIN的接口进行了调整，不再兼容此前的CMake使用方式；通过Makefile.config使用JASMIN的接口保持向后兼容。
3.内部结构：JASMIN 4.0正式版改变了库的打包方式，不再将各个组件作为单独的库文件，而是打包在一起成为一个库文件。文件布局也参照LSB的FHS规范进行了更新。

地址：北京市海淀区花园路6号 邮编：100088 传真：010-61935702
Copyright @ 2014-2022 中物院高性能数值模拟软件中心 京ICP备12047833号-3

[Figure]

微信二维码

[Figure]

**Step 1: Select a JASMIN version**

[Figure]

**Step 2: Fill the application form**

[Figure]

**Step 3: Follow the agreement**

[Figure]

**Step 4: Wait for approval (1~3 working days)**

[Figure]

Step 5: Download after approval (jasmin_JASMIN4-4.0.0-Linux-x86_64.sh)

---

## Author Comment (AC2)

**Author reply to the CEC2 to 'egusphere-2022-510' (Zhang et al., 2022)**

Dear Professor Juan Añel,

Thank you very much for your second comment. Regarding your comment on the status of JASMIN, we have consulted officially with IAPCM. We compile the overall response of OMARE and JASMIN as follows. First, the user of JASMIN (i.e., the authors of this paper) retains all the original rights of the ported code (i.e., OMARE), which we have already published online with our submission. Second, according to the current policy of IAPCM, the current software release of JASMIN is limited to the binary form, and it is licensed through the aforementioned process (in the reply to CEC1).

Regarding your suggestion on the version of JASMIN, we hereby confirm that we have explicit records of the various versions that we used during the development of OMARE. Internally, IAPCM keeps full record of versioning of JASMIN. To produce the results of the study, the version 4.5.0 of JASMIN is used. As stated in our previous reply, it can be applied by emails to: yang_zhang@iapcm.qc.cn or through the online portal at: http://www.caep-scns.ac.cn/JASMIN.php. Detailed guide to the application process is at: https://egusphere.copernicus.org/#AC1.

As a summary of our reply, we want to further emphasize: for our work on OMARE, we did face the dilemma of developing a fully-fledged software infrastructure for GFD (i.e., geophysical fluid dynamics) models (hence adopting a silo-type practice), or utilizing existing, more well-maintained and sophisticated software like JASMIN. In order to suite the various design purpose of OMARE (AMR, parallelization etc.), we chose JASMIN in spite of certain limitations such as its software release status quo. One key factor we weighed in for the decision was that, currently there exists the practice of developing GFD models with commercial (or closed-source) software in GMD journal, or even cases with non-disclosed model code. As pointed out by the CE, this is not an ideal scenario, and we totally agree. Our consideration is that it may be worthy for the modelers (of ocean science or other fields) to explore the possibilities of using third-party middleware software, with JASMIN as the example here. Actually the results we attained provide us (or maybe other model developers as well) the further motivation of seeking for or even developing better infrastructure and tools for GFD in the future. Regarding the CE's suggestion on avoiding the use of this type of software for further study, we will definitely consider it seriously in our future work.

Kind regards,

Shiming Xu and Hengbin An, on behalf of all authors

---

## Author Comment (AC3)

Reply to Referee Comment #2 (RC2):

The authors would like to express sincere thanks to the referee for the invaluable comments. The following are the replies for each point of the comment, together with specific revisions that are made. The original comments are in *green italic* font, and the reply to each of the comments are provided accordingly. Revisions are further highlighted in the revised manuscript in green and marked by **RC2**.

*The manuscript by Zhang et al describes an implementation of a nested regional ocean modeling capability using the parallel computing framework "J Parallel Adaptive Structure Mesh Applications Infrastructure" (JASMIN). They demonstrate the application of the implementation with simulations in an idealized basin double gyre configuration. The presentation is generally clear and the authors provide a realistic assessment of the remaining challenges with this particular implementation as well as for nested ocean modeling generally. My only significant concern with the study is that the entire work is based on a proprietary software stack, making the reproducibility and accessibility of the effort to the broader community questionable.*

**Reply**: the authors thank the referee's comments on the fact that OMARE is based on JASMIN which is a closed-source middleware. During the early design stage of OMARE, we also faced the dilemma of the desired model functionalities (including AMR, automatic restart, parallel I/O, etc.) and the fact that there lacked (and still lacks) openly available   high-performance computing middleware that satisfy these needs. We consider the choice of using JASMIN a worthwhile tryout to leverage the capabilities of traditional models. Furthermore, we do observe various cases in which commercial, closed source software does drive the geophysical modeling community forward, such as NVIDIA CUDA which is the main working horse for many fastest computers in the world.

Therefore, we have open-sourced the whole codebase of OMARE, complying with the rules of GMD, and provided comprehensive guides to applying and using JASMIN. In our opinion, JASMIN is a third-party, multi-purposed software that we utilize in constructing the model, but not the model itself. We definitely welcome the JASMIN developers and its governing bureau on any plan to open-source the platform. Besides, we are also actively searching for open-source alternatives to JASMIN for future developments.

*I suggest below a few additional points and questions that I believe would be helpful to a reader. The grammatical construction (e.g.  singular/plural usage) of the text is a little rough in places and could use some polishing by a technical editor. I have no substantial concerns with the manuscript as written and recommend publication after minor revision.*

**Reply**: the authors have gone through the manuscript to correct small language issues, including those pointed out by the referee.

*Line 31: "Other ensuing practice …" ?? Not right words here*

**Reply**: we change "ensuing practice" into "follow-up modeling practice".

*Line 37: "In the following up part of the paper …" -> In this paper ….*

**Reply**: revised as suggested.

*Line 58: suite -> suit*

**Reply**: corrected.

*Line 70 ASMIN -> JASMIN*

**Reply**: corrected.

*Line 114-119: It would be helpful to actually state the man-months of effort required to complete the refactorization.*

**Reply**: as suggested by the referee, we have added the following sentence for the man-month information: *The whole code refactorization process took about 32 man-months to finish*.

*Line 128: 422 patches … Might be helpful to state where this number comes from .. (# of variables) * (# of grid hierarchy levels) * (???)*

**Reply**: we have reformulated the sentence into: *In total, during the refactorization we have formulated 155 components (i.e., FORTRAN subroutines) and 422 patches (i.e., model variables) in OMARE.*

*Line 131, 133, 175 : Figure 3 should be Figure 1 or Listing 1?*

**Reply**: for all three cases, *Figure 3* is corrected as *Figure 1*.

*Line 180: does level here refer to depth level or grid hierarchy level?*

**Reply**: we confirm that the depth level here refers to the resolution level (or the level of grid hierarchy). The text is modified to contain a more clear description of this information.

*Line 295, Figure 5. Label the figures with the case name or include identifiers in the caption.*

**Reply**: as suggested by the referee, we add the information of the experiment in the figure caption, as follows: _The background and the whole ocean basin is by default in 0.5-deg resolution and marked in cyan, and the region refined to 0.1-deg in each experiment marked in purple._

*Line 296: "Besides, L-M-I covering more area in the subtropical gyre " This seems backwards from the figure. Poor sentence construction.*

**Reply**: we reconstruct the sentence as follows: _Besides, in the subpolar gyre, the refined region to 0.1-deg in L-M-I covers more area than that in L-M-II. On the contrary, in the subtropical gyre, the refined region is smaller in L-M-I than that in L-M-II_.

*Section 3.3: A key conclusion of this section is that the mis-representation of APE->KE transfer in the eddy-parameterized parent-grid leads to boundary forcing biases in the child grid. Later in the paper, the authors allude to the need for a GM-type parameterizations in this regime. Does the implementation support different parameterization methods at different levels of the hierarchy (they show that different parameter values can be used)? An additional case with GM turned on in the 0.5 degree grid would be informative of the suitability of this approach with a scale aware parameterization strategy.*

**Reply**: we confirm that different parameterization schemes could be applied to the different resolution levels in the OMARE system. The mixing schemes already differ among the 3 resolution levels in the numerical experiments of the paper. The experiments based on 0.5-deg relied on a 1st-order Laplacian mixing scheme for lateral mixing of the momentum equation, and the others (with either 0.1-deg or 0.02-deg) adopted a 2nd-order Bi-Laplacian mixing scheme. We plan to further implement other more sophisticated parameterization schemes in the future, in order for a more systematic analysis of the energy cycle and balance of the Double-Gyre system at different resolutions.

*Line 346: "proxies" Not sure this is the correct word here. I believe you mean something more like "metrics of grid quality"*

**Reply**: we change "proxies" to "indicating parameters".

*Line 398: bu -> by*

**Reply**: corrected.

*Line 423: LaTex typo for circle symbol*

**Reply**: corrected.

*Line 424 : "following up part" -> following part*

**Reply**: corrected.

*Line 442: We pick the a -> We pick a*

**Reply**: corrected.

*Line 474: This is indicated by that both … -> This is indicated by the fact that both …*

**Reply**: revised as indicated by the referee.

*Line 505-514: Could alternatively provide an estimate of the man-months of effort here.*

**Reply**: the information of man-month is added as follows: *The refactorization process in total involves about 32 man-months to finish*.

*Line 529: (Section on refinement criteria) : Some description of how the current implenmexation aggregates points into individual patches would be helpful.*

**Reply**: we hereby confirm with the referee that the aggregation of model points into patches are mainly carried out by the middleware software of JASMIN. For the refined region, the user can specify the granularity of the refined region, which indirectly controls

the patch size for the refined region. For example, we use the granularity parameter of 3, and given the refinement ratio of 5, the refined region is then consisted of regions of 15x15 in size on the refined level (or 3x3 grid points on the coarse level), and the resulting patch size is 15x15 as well. This granularity parameter affects the computational performance and should be treated as a trade-off for model tuning. We consider it a technical detail and hence omitted it from the manuscript. As suggested by the referee, we add the following concise description in this section: _Specifically, a prescribed parameter for refinement granularity can be used to control the patch size on the refined region, ensuring both the full coverage to these marked grid cells and patch sizes which affects computational performance._

_Line 597 ( Section on realistic cases): Some discussion of challenges expected with realistic topography and coastlines would be helpful. Would topography or the coastline be refined along with the grid? How would mismatches in land-ocean boundaries across hierarchy levels at the lateral boundary be handled?_

**Reply**: as suggested by the referee, we have included some discussion of potential issues and challenges with realistic bathymetry, including the choice of refinement, as well as the cross-boundary consistency of model bathymetry. The added sentences are: "_Specifically, spatial refinement can be carried out in key regions with bathymetric features, such as land-sea boundaries, continental shelves, sea mounts. Due to the different bathymetry across the resolutions in OMARE, the model status on the coarse grid contains inherent inconsistencies for the refined region. Therefore, after spatial and temporal interpolation, the lateral boundary conditions to the refined region need to be modified accordingly, in order to reduce any potential physical and numerical issues_".

_Other: I presume some data on scalability and performance will be presented in the companion manuscript, but a very brief statement about this would be helpful to make the present manuscript self-contained._

**Reply**: we have revised the last sentence of the first paragraph in Summary, in order to include the brief information for the planned accompanying paper. The revision is as follows: "_Another planned paper (part 2) will further introduce the computational aspects of OMARE, including the scalability and computability of OMARE, with a particular focus on AMR and its role in improving the computational efficiency of high-resolution simulations_."

---

## Author Comment (AC4)

**Reply to Referee Comment #1 (RC1):**

The authors would like to thank the referee for the invaluable comments and suggestions. The following are the replies for each point of the comment, together with specific revisions that are made. The original comments are in *blue italic* font and listed in paragraphs, with our reply following each paragraph separately. The revisions are also highlighted in the revised manuscript in blue and marked by **RC1**.

I find the approach proposed in the manuscript to be of interest for the journal. It follows the now discussed road of separation of concerns, whereby the code part dealing with numerical algorithms and the part dealing with the infrastructure (mesh, parallelization) are separated and treated in different ways. However, the manuscript in its present form misses the goal: one expects that the material of the manuscript describing a modeling approach will be sufficient for a reader to learn how to use the approach. I do not see that this goal is reached.

**Reply:** the authors thank the referee's comment on the relevance of our contribution to the journal. Regarding the comment on the main purpose of the manuscript, we would like to make the following clarifications and corresponding revisions to the manuscript.

First, we would like to clarify that we have multiple purposes for the manuscript: (1) we demonstrate that by refactoring existing model (here NEMO) with JASMIN we can attain extra functionality, while breaking the `silo'-type model development in which modelers make (almost) everything by themselves; (2) the resulting model OMARE is capable to simulate realistic ocean processes with the AMR function working as intended; (3) the introduction of the actual refactorization process, using NEMO and JASMIN as an example. We consider OMARE a worthy try in (re-)constructing the model with a software middleware and satisfactory results are achieved. We agree with the referee that the technical details of how to refactorize the model is an important aspect, but we consider it one of these goals. In response to the referee's suggestion, we provide a technical guide as supplementary material for using JASMIN and compiling/running OMARE (see also the reply to the next paragraph). The text referring to the new technical guide is also added to the revised version of the manuscript.

In order to better explain the overall design idea of OMARE, we intend to add the following figure of the model structure of OMARE and differentiate it with NEMO. The figure, together with the added paragraph to the manuscript is now included in the beginning of Sec. 2 of the revised version of the manuscript.

NEMO, as a typical ocean model, consists of a layered structure. Like many models, it relies on an intermediate layer that contains both a self-developed parallelization software solution, and third-party add-on's (AGRIF, XIOS). As a consequence, certain limitations exist, including limited flexibility in parallelization and adaptivity, especially given that these issues are inherently intertwined. OMARE, on the contrary, relies on JASMIN for managing the parallelization, adaptive refinement, and parallel I/O. Therefore, no extra effort is spent on designing/building the intermediate layer

specifically for the model, hence no software `silo' is constructed. More importantly, the model does not suffer from the aforementioned limitations.

| Į Į           | Numerical & Physical Layer                                                                                               | Numerical & Physical Layer                                                                                                                                        |
|---------------|--------------------------------------------------------------------------------------------------------------------------|-------------------------------------------------------------------------------------------------------------------------------------------------------------------|
| Applicat      | Forcings Active Parameterizations •••• Baroclinic Barotropic Dynamics Dynamics                                           | Forcings Active Tracers Parameterizations ••• Baroclinic Dynamics Dynamics                                                                                        |
| ſ             | Parallel-support Layer                                                                                                   | JASMIN                                                                                                                                                            |
| oftwares      | Domain
Decomposition
Process
Mapping
Wrapper to Communication
(based on MPI)                              | Parallel Computing Support Domain Decomposition Parallel Process Mapping API to Communication (based on MPI)                                                      |
| Middleware So | AGRIF
Grid Hierarchy
Management
Inter-level
Exchange                                                         | Embedded Grids
Grid Hierarchy
Management
Inter-level
Exchange                                                                                         |
| nfrastructure | High-performance Cluster         MPI Environment           Interconnect         Storage           Network         System | High-performance Cluster         MPI Environment           Interconnect
Network         Storage
System         Memory Hierarchy         Operating
System |

Figure 1. Model structure and abstraction layers of NEMO (left) and OMARE (right).

Of course, there are also limitations associated with OMARE, such as the lack of direct support for accelerator architectures (e.g., GPUs) in JASMIN. Besides, there are potential usability issues due to the current binary release form of JASMIN, which is an issue also raised by other editors. We are aware that these detailed issues should be addressed during future development of OMARE.

The text on lines 165 - 190 shortly describes how the JASMIN is involved, but I doubt a reader can get any understanding of what and why is done. Moreover, it is not at all clear how to use JASMIN in conjunction with the updated code. How the JASMIN environment can be installed, how code is compiled, etc. The description should be essentially extended and be such that those who are willing to follow author's approach can do it.

**Reply:** we agree with the referee on the importance of the technical aspects (of how JASMIN and the application work together). We do consider the detailed content will be too much to be included in the main part of the paper, considering the overall purpose of the work, as well as the limited relevance of these details only to model developers. Therefore, we supply a concise manual for the installation of JASMIN and the compilation and running of OMARE in the supplementary. The text here is also revised with the addition of the reference to this manual.

The manuscript devotes more than a half of its volume to the description of simple test configuration, going into too much detail, which is hardly optimal. The test case remains the test case, and one can only learn that the approach proposed by the authors is working, yet not without drawbacks related to one-way nesting (the development of errors on fine-coarse boundary). I do not think that this test case is well suited to demonstrate the need of adaptivity. Figure 11 shows clearly that small-scale turbulence occupies 3/4 domain on full mesh, and it occupies only pieces where the resolution is refined in b and c. It is different from the initial phases in Fig. 9 and 10, but 5 or 20 days is a too short time for turbulence to equilibrate, and this transient phase is of no interest (it depends on coarse initial conditions, and does not model any reality). So the conclusion here is that dynamic adaptivity is an interesting, but perhaps not very needed possibility as concerns eddying flows. Static refinement might be doing the work, and one will take a decision where to resolve based on one interest. I foresee, however, one direction, where dynamic refinement still might be of interest -- the simulations of seasonal course of variability. Submesoscale eddies might be suppressed in warm seasons, and a coarser mesh will be sufficient for mesoscale. My recommendation are to make the experimental part more compact. One can hardly learn anything from detailed description of particular eddy features or the comparison of transects (Fig. 13, 14), and there is very little sense in Fig. 9 and 10.

**Reply:** we would like to thank the referee for the comment on the test case. We would like to make the following replies and revisions to the manuscript.

**First, on the choice of Double-Gyre testcase**. The Double-Gyre testcase is chosen as a testbed for the major functionalities of OMARE. The case is of intermediate complexity: (1) the model physics is complete, making it capable to produce realistic three-dimensional large-scale ocean processes; (2) it produces an idealized Western Boundary Current system and in particular, the seasonality of submesoscale processes; (3) the testcase omits realistic (or any) bathymetry, therefore avoiding the complex issues such as inconsistent bathymetry between the non-refined and the refined regions. We agree with the referee that the Double-Gyre is a choice among many possible ones. And we chose it mainly because that the aforementioned 3 characteristics. Similar Double-Gyre cases or even more complex idealized cases such as NeverWorld2 (Marques et al., 2022) are also commonly used in the community for model benchmarking or the study of certain processes.

**Second, on the demonstration of the need for adaptivity**. We agree with the referee that small-scale turbulence is prevalent in the basin for full-basin 0.02-deg experiments, which is actually expected especially during winter. Here we would like to emphasize our perspective on adaptive refinement: where & when we need adaptivity and AMR is an open issue, and the purpose of OMARE is to provide a framework that supports various possible AMR scenarios.

In the Double-Gyre case, the major simulated features include the WBC, the lateral boundaries of the basin, and the associated mesoscale-submesoscale processes. The refinement criteria are designed to capture the kinetically active and submesoscale

processes, and hence based on surface velocity and surface relative vorticity. The two AMR cases have already been shown to capture a majority of the basin's (surface) kinetic energy (Fig. 8), which directly indicates the validity of AMR.

A further proof is provided below in Fig. 2 for the vertical motion induced by submesoscale flow. The enhanced vertical motion and the associated heat (and other tracer) transport is a key characteristics of the submesoscale processes (Taylor & Thompson, 2022). The following figure shows that: (1) the strong vertical motion is strongly concentrated at WBC and some other regions (i.e., ocean fronts, basin's boundary), and (2) the two AMR cases capture these key regions for strong vertical motion. A further investigation of the PDF of vertical speed in Fig. 3 confirms that the strong vertical motion in AMR experiments matches closely with the full-field 0.02-deg experiment.

---

## Author Comment (AC5)

**Authors' Final Response to comments to egusphere-2022-510, entitled "Ocean Modeling with Adaptive REsolution (OMARE, version 1.0) – Refactoring NEMO model (version 4.0.1) with the parallel computing framework of JASMIN. Part 1: adaptive grid refinement in an idealized double-gyre case"**

The authors would like to thank the editors and the two referees for the invaluable comments and suggestions, which helps us a lot in improving both our own understanding of the context and the manuscript. We have replied to the editors' comments and the referee's comments accordingly. In this final response we make a summary to these replies and attach the specific replies to the referee's comments (RC1 and RC2). Specifically, the reply to RC1 occupies the 3rd to the 8th page, and the reply to RC2 occupies the 9th to 13th page. The manuscript is revised accordingly, and the marked-up version provided.

In this final response, we would like to first re-emphasis our main goal of our work of OMARE, which is a refactorization of NEMO with the high-performance middleware software of JASMIN. Our main goal is to demonstrate that: advanced model functionalities, such as adaptive refinement and automatic parallelization can be achieved for current models with such a refactorization practice. In most models, the model parallelization is usually carried out by the model development team, which in effect results in a `silo'-type software and undermines long-term development. But functionalities such as adaptive refinement (AMR) is a closely related model design to domain decomposition and parallelization, and it cannot be easily implemented and usually beyond the capacity of many model development teams. By utilizing JASMIN or similar general-purpose middleware software, the model developer is alleviated from these development efforts, and the model functionality can also be improved. Furthermore, by adaptive refinement down to 0.02-deg resolution, OMARE simulates turbulent ocean processes and captures the dynamically changing mesoscale and submesoscale on the Western Boundary Current system.

Second, we would like to summarize our reply to the status of JASMIN. We have provided several proofs and efforts alongside the manuscript to improve the usability of OMARE and JASMIN. (A) We have provided the full codebase of OMARE, consisting of over 40000 new lines of code, which complies fully with the GMD protocol. (B) We have provided several ways, including a step-by-step instruction, for the application of the proper version of JASMIN, which is currently released in binary form from its developing team. (C) We provide a brief guide for installing JASMIN, as well as compiling and running OMARE. These guides are provided as either Author Comment (https://editor.copernicus.org/#AC4) or the supplementary of the revised manuscript. JASMIN is independently developed and well managed at the Institute of Applied Physics and Computational Mathematic (IAPCM) and IAPCM retains all the copyright to it. We consider JASMIN an external, infrastructural software that we build OMARE with, but NOT the model itself. And we are fully aware of the various cases in the community that the infrastructural software is commercial (e.g., NVIDIA CUDA) and therefore not accessible in the open-source sense, and even the source of the model itself is NOT released (due to policy limitations of the governing body). Despite the practical limitations of using JASMIN, we hope that our practice could contribute a worthy try for the long-term development of geofluid models.

Third, we have finished the replies to both referees' general and specific comments, and attached them in this reply as well. The general comments from the referees mainly concerns the purpose of our work, the analysis of the experiments and the layout of the related text, as well as the usability of JASMIN (which we addressed above). We have finished item-to-item replies, and made revisions to the manuscript. The marked-up version of the revised manuscript is also provided, with specific notes to either of the referees' comments (RC1 or RC2).

We would like to express our sincere thanks again to the two editors and the two referees for their help to improve our manuscript. We would also like to send them our greetings for the upcoming holiday season and the new year of 2023!

Shiming Xu, on behalf of all authors

**Reply to Referee Comment #1 (RC1):**

The authors would like to thank the referee for the invaluable comments and suggestions. The following are the replies for each point of the comment, together with specific revisions that are made. The original comments are in *blue italic* font and listed in paragraphs, with our reply following each paragraph separately. The revisions are also highlighted in the revised manuscript in blue and marked by **RC1**.

I find the approach proposed in the manuscript to be of interest for the journal. It follows the now discussed road of separation of concerns, whereby the code part dealing with numerical algorithms and the part dealing with the infrastructure (mesh, parallelization) are separated and treated in different ways. However, the manuscript in its present form misses the goal: one expects that the material of the manuscript describing a modeling approach will be sufficient for a reader to learn how to use the approach. I do not see that this goal is reached.

**Reply:** the authors thank the referee's comment on the relevance of our contribution to the journal. Regarding the comment on the main purpose of the manuscript, we would like to make the following clarifications and corresponding revisions to the manuscript.

First, we would like to clarify that we have multiple purposes for the manuscript: (1) we demonstrate that by refactoring existing model (here NEMO) with JASMIN we can attain extra functionality, while breaking the `silo'-type model development in which modelers make (almost) everything by themselves; (2) the resulting model OMARE is capable to simulate realistic ocean processes with the AMR function working as intended; (3) the introduction of the actual refactorization process, using NEMO and JASMIN as an example. We consider OMARE a worthy try in (re-)constructing the model with a software middleware and satisfactory results are achieved. We agree with the referee that the technical details of how to refactorize the model is an important aspect, but we consider it one of these goals. In response to the referee's suggestion, we provide a technical guide as supplementary material for using JASMIN and compiling/running OMARE (see also the reply to the next paragraph). The text referring to the new technical guide is also added to the revised version of the manuscript.

In order to better explain the overall design idea of OMARE, we intend to add the following figure of the model structure of OMARE and differentiate it with NEMO. The figure, together with the added paragraph to the manuscript is now included in the beginning of Sec. 2 of the revised version of the manuscript.

NEMO, as a typical ocean model, consists of a layered structure. Like many models, it relies on an intermediate layer that contains both a self-developed parallelization software solution, and third-party add-on's (AGRIF, XIOS). As a consequence, certain limitations exist, including limited flexibility in parallelization and adaptivity, especially given that these issues are inherently intertwined. OMARE, on the contrary, relies on JASMIN for managing the parallelization, adaptive refinement, and parallel I/O. Therefore, no extra effort is spent on designing/building the intermediate layer

specifically for the model, hence no software `silo' is constructed. More importantly, the model does not suffer from the aforementioned limitations.

|               | Numerical & Physical Layer                                                                                               | Numerical & Physical Layer                                                                                                                     |
|---------------|--------------------------------------------------------------------------------------------------------------------------|------------------------------------------------------------------------------------------------------------------------------------------------|
| Applicat      | Forcings Active Parameterizations ••• Baroclinic Dynamics Dynamics                                                       | Forcings Active Tracers Parameterizations ••• Baroclinic Dynamics Dynamics                                                                     |
| ſ             | Parallel-support Layer                                                                                                   | JASMIN                                                                                                                                         |
| oftwares      | Domain
Decomposition
Mapping
Parallel
Process
Mapping
Wrapper to Communication
(based on MPI)       | Parallel Computing Support           Domain         Parallel
Process         API to Communication
(based on MPI)                         |
| MiddleWare S  | AGRIF
Grid Hierarchy
Management
Inter-level
Exchange VO Support NetCDF XIOS                                  | Embedded Grids
Grid Hierarchy
Management
Inter-level
Exchange
Utilities
Performance
Profiling
HDF5 VO
Visualization |
| ntrastructure | High-performance Cluster         MPI Environment           Interconnect         Storage           Network         System | High-performance Cluster         MPI Environment           Interconnect         Storage           Network         System                       |

Figure 1. Model structure and abstraction layers of NEMO (left) and OMARE (right).

Of course, there are also limitations associated with OMARE, such as the lack of direct support for accelerator architectures (e.g., GPUs) in JASMIN. Besides, there are potential usability issues due to the current binary release form of JASMIN, which is an issue also raised by other editors. We are aware that these detailed issues should be addressed during future development of OMARE.

The text on lines 165 - 190 shortly describes how the JASMIN is involved, but I doubt a reader can get any understanding of what and why is done. Moreover, it is not at all clear how to use JASMIN in conjunction with the updated code. How the JASMIN environment can be installed, how code is compiled, etc. The description should be essentially extended and be such that those who are willing to follow author's approach can do it.

**Reply:** we agree with the referee on the importance of the technical aspects (of how JASMIN and the application work together). We do consider the detailed content will be too much to be included in the main part of the paper, considering the overall purpose of the work, as well as the limited relevance of these details only to model developers. Therefore, we supply a concise manual for the installation of JASMIN and the compilation and running of OMARE in the supplementary. The text here is also revised with the addition of the reference to this manual.

The manuscript devotes more than a half of its volume to the description of simple test configuration, going into too much detail, which is hardly optimal. The test case remains the test case, and one can only learn that the approach proposed by the authors is working, yet not without drawbacks related to one-way nesting (the development of errors on fine-coarse boundary). I do not think that this test case is well suited to demonstrate the need of adaptivity. Figure 11 shows clearly that small-scale turbulence occupies 3/4 domain on full mesh, and it occupies only pieces where the resolution is refined in b and c. It is different from the initial phases in Fig. 9 and 10, but 5 or 20 days is a too short time for turbulence to equilibrate, and this transient phase is of no interest (it depends on coarse initial conditions, and does not model any reality). So the conclusion here is that dynamic adaptivity is an interesting, but perhaps not very needed possibility as concerns eddying flows. Static refinement might be doing the work, and one will take a decision where to resolve based on one interest. I foresee, however, one direction, where dynamic refinement still might be of interest -- the simulations of seasonal course of variability. Submesoscale eddies might be suppressed in warm seasons, and a coarser mesh will be sufficient for mesoscale. My recommendation are to make the experimental part more compact. One can hardly learn anything from detailed description of particular eddy features or the comparison of transects (Fig. 13, 14), and there is very little sense in Fig. 9 and 10.

**Reply:** we would like to thank the referee for the comment on the test case. We would like to make the following replies and revisions to the manuscript.

**First, on the choice of Double-Gyre testcase**. The Double-Gyre testcase is chosen as a testbed for the major functionalities of OMARE. The case is of intermediate complexity: (1) the model physics is complete, making it capable to produce realistic three-dimensional large-scale ocean processes; (2) it produces an idealized Western Boundary Current system and in particular, the seasonality of submesoscale processes; (3) the testcase omits realistic (or any) bathymetry, therefore avoiding the complex issues such as inconsistent bathymetry between the non-refined and the refined regions. We agree with the referee that the Double-Gyre is a choice among many possible ones. And we chose it mainly because that the aforementioned 3 characteristics. Similar Double-Gyre cases or even more complex idealized cases such as NeverWorld2 (Marques et al., 2022) are also commonly used in the community for model benchmarking or the study of certain processes.

**Second, on the demonstration of the need for adaptivity**. We agree with the referee that small-scale turbulence is prevalent in the basin for full-basin 0.02-deg experiments, which is actually expected especially during winter. Here we would like to emphasize our perspective on adaptive refinement: where & when we need adaptivity and AMR is an open issue, and the purpose of OMARE is to provide a framework that supports various possible AMR scenarios.

In the Double-Gyre case, the major simulated features include the WBC, the lateral boundaries of the basin, and the associated mesoscale-submesoscale processes. The refinement criteria are designed to capture the kinetically active and submesoscale

processes, and hence based on surface velocity and surface relative vorticity. The two AMR cases have already been shown to capture a majority of the basin's (surface) kinetic energy (Fig. 8), which directly indicates the validity of AMR.

A further proof is provided below in Fig. 2 for the vertical motion induced by submesoscale flow. The enhanced vertical motion and the associated heat (and other tracer) transport is a key characteristics of the submesoscale processes (Taylor & Thompson, 2022). The following figure shows that: (1) the strong vertical motion is strongly concentrated at WBC and some other regions (i.e., ocean fronts, basin's boundary), and (2) the two AMR cases capture these key regions for strong vertical motion. A further investigation of the PDF of vertical speed in Fig. 3 confirms that the strong vertical motion in AMR experiments matches closely with the full-field 0.02-deg experiment.

---

## Author Response (AR2)

**Reply to Referee Comment #3 (RC3):**

The authors would like to thank the referee for the follow-up comments to the revised version of the manuscript. We would like to make the following replies, specific to each comment. The original comments are in *purple italic* font and listed in paragraphs, with our reply following each paragraph separately. The revisions are also highlighted in the revised manuscript in *purple* and marked by *RC3*.

**The manuscript has improved, and I am almost satisfied with it except for several small issues.**

1. The main point of section 1.1 is that other existing solutions (AGRIF or unstructured meshes) do not have the capability adaptive grid refinement. I do not think this is the main point. There are unstructured-grid models that use adaptive mesh refinement, especially as concerns tsunami applications. However, in practical terms it would be very difficult to propose a situation when adaptive mesh refinement is needed in realistic ocean simulations. One generally knows where eddies are present, and static refinement is sufficient in most cases. Besides, remeshing destroys geostropic and other balances, which is not necessarily a welcome addition. This is the main reason why adaptive meshes are not used in ocean modeling, different from computational fluid dynamics.

If one is following the development of a shock wave, mesh adaptivity is very helpful. The discussion of missing adaptivity is therefore misleading. In my opinion, the interesting part of the manuscript is that the authors present a use case of an approach based on the separation of concerns, where the parallel infrastructure is separated from the numerical part, and this should serve as the main motivation.

**Reply**: the authors fully agree with the referee that there exist ocean modeling activities that utilize adaptive refinement. Therefore, we would like to revise the text to state clearly that OMARE is targeted at the adaptive refinement for the modeling of the ocean's general circulation. Hereby we would also like to make clarification of our understanding of the certain scenarios in ocean modeling that do require adaptive refinement. One particular scenario is the tracking of certain features that are of interest by dynamically changing. For example, the mesoscale eddies shed from the Agulhas Retroflection propagate into the southern Atlantic. By dynamically tracking these eddies with refined resolution (e.g., 10km and finer) would help better understand the salinity transport and climatic impact of these eddies, as well as their dissipation and interaction with bathymetry.

Another clarification is about potentially breaking the geostrophic balance during the simulation. Even with dynamic refinement, the model still carries out full simulation on the coarse resolution. Therefore, the geostrophic balance on the coarse resolution level is not violated. Indeed, across the resolution boundaries, there exist potential inconsistencies during long-term runs, and we are working on improving the model to feedback (or, in a sense, improve) the status on the coarse resolution. And this would alleviate the aforementioned inconsistencies, and it will be an add-on functionality of OMARE's next version.

The authors thank the referee for expressing the interest in the separation of model development concerns. We do consider this practice of value to the community, since it alleviates the model developers from the details of parallel computing, while maintaining good overall computational performance. According to the referee's comment, we revise the Section 1 accordingly. (1) We add the specific statement of the adaptive refinement that better clarify our motivation, as well as the role of Double-Gyre case which is to demonstrate this functionality. (2) We better introduce our motivation of refactoring/constructing models with JASMIN-like middlewares that better separate computation from model physics/numerics and alleviate the modeler's efforts.

2. I have already written previously that the extensive analysis of the double-gyre case is of very limited sense. The metric of KE is not very much telling because the author do not control energy input in their model. I would recommend to further shorten the description. As concerns diagnostics, it would be advisable to look at mean eddy kinetic energy distribution in space (for 0.5 - 0.1 degree experiments). In the adaptive refinement case one can hardly learn anything from multiple panels after 5 20 or 50 days. These time intervals are too short for studying real ocean. The authors raise a question on matching dynamics between coarse and fine subdomains, but only in passing. Do one needs, for example, to adjust dissipation at the boundary?

**Reply**: the authors agree with the referee that the energy budget of the Double-Gyre should be better accounted for. We do confirm with the referee that the energy input into the system, both dynamic and thermodynamic, is indeed constrained. As evident from the manuscript, the energy cycle of the 0.5-deg run and that of the 0.1-deg run are very different, especially given that no mesoscale turbulence and energy reservoir is present in the 0.5-deg run.

We confirm that we did investigate the eddy kinetic energy of the 0.1-deg run. The WBC and its extension is the 'hot spot' of kinetic energy. We now provide the figure as an extra supplementary to the revised manuscript, and provided below as well.

According to the referee's suggestion on the 5-day, 20-day and 50-day results of the 0.02-deg experiments, we further shorten and simplify the text. We intend to keep the figures for the content to be indicative of the Adaptive Refinement working as we expected. For the inter-boundary exchange, we first plan to better match the coarse-fine grid status by updating the coarse grid status by that of the fine grid for future work of OMARE (which we mentioned in the final section of the manuscript). Meanwhile, numerical dissipation may be needed after synchronizing the model status, which we plan to carry out further.

Figure. Annual mean eddy kinetic energy  $(m^2/s^2)$  for 0.1-deg simulation based on daily instantaneous model status.

3. There are still too many grammatical errors (singular-plural is the most frequent one), and some phrases have to be edited. I have not compiled the list, but please read the text very carefully, improvements are needed in many places.

**Reply**: we have gone thoroughly through the manuscript to correct for the errors. These revisions are also highlighted. We apologize for any inconveniences that are caused to the referee.

Below I specifically mention some places.

line 47 "First ..." --- and where is "Second..."?

Reply: we revised the second sentence to add the missing "Second".

line 57 "Similarly ... " -- it is not an English sentence

**Reply**: we revised the sentence from "Similarly, with MPAS, grid generators ..." to "There is similar practice for MPAS, which utilize grid generators ..."

*Line 58 "Although existing models (which are based on orthogonal grids) ... " -- What are you willing to say? Do you mean structured-mesh models?*

**Reply**: the sentence is revised as: "Although existing structured-grid based models can no longer ..."

**Line 60 "(1) the model grids ...." --- there is reason to avoid this**

**Reply**: we have revised the sentence by segmenting it into 4 individual ones, as follows: "However, there are certain limitations. First, the model grids cannot change arbitrarily with time, hence limited 'adaptivity' and 'flexibility'. Second, scale-aware parameterization schemes should be developed to accommodate gradual change of model grid resolution. Third, due to CFL limitations, the time step is usually controlled by the smallest grid cell size, resulting in extra computational cost."

line 64 "Furthermore, there is no ocean model ..." --- because nobody ever shown that moving grids are needed for ocean simulations. In neXtSim they are trying to preserve the structure of linear kinematic features, but why it should be used in the large-scale ocean?

**Reply**: we would like to clarify that to our understanding, we have the following motivation for using adaptive refinement for modeling the ocean's general circulation. First, the tracking of dynamic processes requires adaptive refinement, such as mesoscale eddies, ocean fronts, etc. Second, large-scale features such as polar sea ice and submesoscale processes are of clear seasonal characteristics, and they serve as potential candidates for adaptive refinement. Third, certain non-scientific ocean applications such as the tracking of arbitrarily moving objects/Lagrangian points. We would like to re-emphasize that in this work, we use Double-Gyre test case to demonstrate the AMR capability of OMARE, which is shown to simulate reasonable ocean processes. The possible use of OMARE and its AMR function is still subjected to specific simulation scenarios.

**line 80 "tripolar boundaries" --- what is tripolar boundaries?**

**Reply**: tripolar grids are commonly used for many high-resolution global ocean models as model grids. The tripolar boundary is the grid's northest boundary, which connects the 'eastern' hemisphere and the 'western' hemisphere of the model grid. We have revised the sentence and include the necessary reference to Murray (1996).

*line 95 "including first-order and second-order viscosity/diffusion" --- What do you mean? Do you mean harmonic and biharmonic viscosity and diffusion? They can be of arbitrary order of accuracy.*

**Reply**: we have revised the manuscript accordingly. Indeed we mean harmonic and biharmonic viscosity/diffusion schemes.

**Fig. 1 Please provide an extended caption. I can hardly see what is the intention of this figure.**

**Reply**: we have revised the figure caption to include relevant information of the figure. The main purpose of this added figure (in revision 1) is to contrast the model structure of NEMO and OMARE (i.e., after refactorization).

**Fig. 3 Same comment. I do not see what I should conclude from this figure.**

**Reply**: we have revised the figure caption to include relevant information of the figure. The main purpose is to demonstrate the typical time integration in OMARE, including the adaptive refinement cycle, which is a sequence of baroclinic steps of the coarsest resolution during simulation.

*line 296 What is 'kinetic'? Do you mean kinetic energy? This issue is present in several places.*

**Reply**: we revised "the surface ocean kinetics" into "the mean surface kinetic energy". Other places of misuse are also corrected.

**Section 3.3 and further -- use roman font for units**

Reply: we have revised the manuscript accordingly, including the text and Figure 7.

line 318 PE is seemingly the potential energy, which is not stated. I do not see the definition of PE, and from what I can reconstruct from the text, the authors identify the PE with the difference in sea surface height which is an error. One is interested in equilibrated regimes and not in transient phenomena in toy models. Why all this discussion is needed?

**Reply**: we have revised to include the precise statement, which is the "mean potential energy (PE)". We have now added the statement at its first use. At 0.5-deg, the model simulates systematically higher mean PE than that at 0.1-deg. As a future study, a holistic analysis of the energy reservoir and the transfer and comparison between 0.5-deg and 0.1-deg experiments are planned. We would also like to clarify that the comparison of 0.5-deg and 0.1-deg experiments, including the transitional phase, is to contrast the systematic difference of these full-field and regionally-refined simulations.